# Three-dimensional ultrasound matrix imaging

**Flavien Bureau[1], Justine Robin[1,2], Arthur Le Ber [1], William Lambert [1,3], Mathias Fink [1] & Alexandre Aubry [1]**

Matrix imaging paves the way towards a next revolution in wave physics. Based on the response matrix recorded between a set of sensors, it enables an optimized compensation of aberration phenomena and multiple scattering events that usually drastically hinder the focusing process in heterogeneous media. Although it gave rise to spectacular results in optical microscopy or seismic imaging, the success of matrix imaging has been so far relatively limited with ultrasonic waves because wave control is generally only performed with a linear array of transducers. In this paper, we extend ultrasound matrix imaging to a 3D geometry. Switching from a 1D to a 2D probe enables a much sharper estimation of the transmission matrix that links each transducer and each medium voxel. Here, we first present an experimental proof of concept on a tissue-mimicking phantom through ex-vivo tissues and then, show the potential of 3D matrix imaging for transcranial applications.

The resolution of a wave imaging system can be defined as the ability to discern small details of an object. In conventional imaging, this resolution cannot overcome the diffraction limit of a half wavelength and may be further limited by the maximum collection angle of the imaging device. However, even with a perfect imaging system, the image quality is affected by the inhomogeneities of the propagation medium. Large-scale spatial variations of the wave velocity introduce aberrations as the wave passes through the medium of interest. Strong concentration of scatterers also induces multiple scattering events that randomize the directions of wave propagation, leading to a strong degradation of the image resolution and contrast. Such problems are encountered in all domains of wave physics, in particular for the inspection of biological tissues, whether it be by ultrasound imaging[1] or optical microscopy[2], or for the probing of natural resources or deep structure of the Earth's crust with seismic waves[3].

To mitigate those problems, the concept of adaptive focusing has been adapted from astronomy where it was developed decades ago[4,5]. Ultrasound imaging employs array of transducers that allow to control and record the amplitude and phase of broadband wave fields. Wavefront distortions can be compensated for by adjusting the time delays added to each emitted and/or detected signal in order to focus ultrasonic waves at a certain position inside the medium[6–9]. The estimation of those time delays implies an iterative time-consuming focusing process that should be ideally repeated for each point in the field of view[10,11]. Such a complex adaptive focusing scheme cannot be implemented in real time since it is extremely sensitive to motion[12] whether induced by the operator holding the probe or by the movement of tissues.

Fortunately, this tedious process can now be performed in postprocessing[13,14] thanks to the tremendous progress made in terms of computational power and memory capacity during the last decade. To optimize the focusing process and image formation, a matrix formalism can be fruitful[15–18]. Indeed, once the reflection matrix **R** of the impulse responses between each transducer is known, any physical experiment can be achieved numerically, either in a causal or anti-causal way, for any incident beam and as many times as desired. More specifically, assuming that the medium remains fixed during the acquisition, multi-scale analysis of the wave distortions can be performed to build an estimator of the transmission matrix **T** between each transducer of the probe and each voxel inside the medium[19]. Once the **T**-matrix is known, a local compensation of aberrations can be performed for each voxel, thereby providing a confocal image of the medium with a close-to-ideal resolution and an optimized contrast everywhere.

[1]Institut Langevin, ESPCI Paris, PSL University, CNRS, 75005 Paris, France. [2]Physics for Medicine, ESPCI Paris, PSL University, INSERM, CNRS, Paris, France. [3]Hologic / SuperSonic Imagine, 135 Rue Emilien Gautier, 13290 Aix-en-Provence, France. ✉e-mail: Alexandre.Aubry@espci.fr

Although it gave rise to striking results in optical microscopy[20–24] or seismic imaging[25,26], the experimental demonstration of matrix imaging has been, so far, less spectacular with ultrasonic waves[17,18,27,28]. Indeed, the first proof-of-concept experiments employed a linear array of transducers. Yet, aberrations in the human body are 3D-distributed and a 1D control of the wave field is not sufficient for a fine compensation of wave distortions as already shown by previous works[29–32]. Moreover, 2D imaging limits the density of independent speckle grains which controls the spatial resolution of the **T**-matrix estimator[28].

In this work, we extend the ultrasound matrix imaging (UMI) framework to 3D using a fully populated matrix array of transducers[33–35]. The overall method is first validated by means of a well-controlled experiment combining ex-vivo pork tissues as aberrating layer on top of a tissue-mimicking phantom. 3D UMI is then applied to a head phantom whose skull induces a strong attenuation, aberration, and multiple scattering of the ultrasonic wave field, phenomena that UMI can quantify independently of each other[1,19]. Inspired by the CLASS method developed in optical microscopy[20,22], aberrations are here compensated by a novel iterative phase reversal algorithm more efficient for 3D UMI than a singular value

decomposition[16–18]. In contrast with previous works, the convergence of this algorithm is ensured by investigating the spatial reciprocity between the **T**-matrices in transmission and reception. Throughout the paper, we will compare the gain in terms of resolution and contrast provided by 3D UMI with respect to its 2D counterpart. In particular, we will demonstrate how 3D UMI can be a powerful tool for optimizing the focusing process inside the brain through the skull.

## Results

### Beamforming the reflection matrix on a focused basis

3D UMI starts with the acquisition of the reflection matrix (see Methods) by means of a 2D array of transducers [32 × 32 elements, see Fig. 1a, b]. It was performed first on a tissue-mimicking phantom with nylon rods through a layer of pork tissue of fat and muscle (obtained from a chop rib piece), acting as an aberrating layer [Fig. 2a], and then on a head phantom including brain and skull-mimicking tissue, to reproduce transcranial imaging (see below). In the first experiment, the reflection matrix $\mathbf{R_{uu}}(t)$ is recorded in the transducer basis [Fig. 1a, c], i.e. by acquiring the impulse responses, $R(\mathbf{u_{in}}, \mathbf{u_{out}}, t)$, between each transducer ($\mathbf{u}$) of the probe. In the head phantom experiment, skull

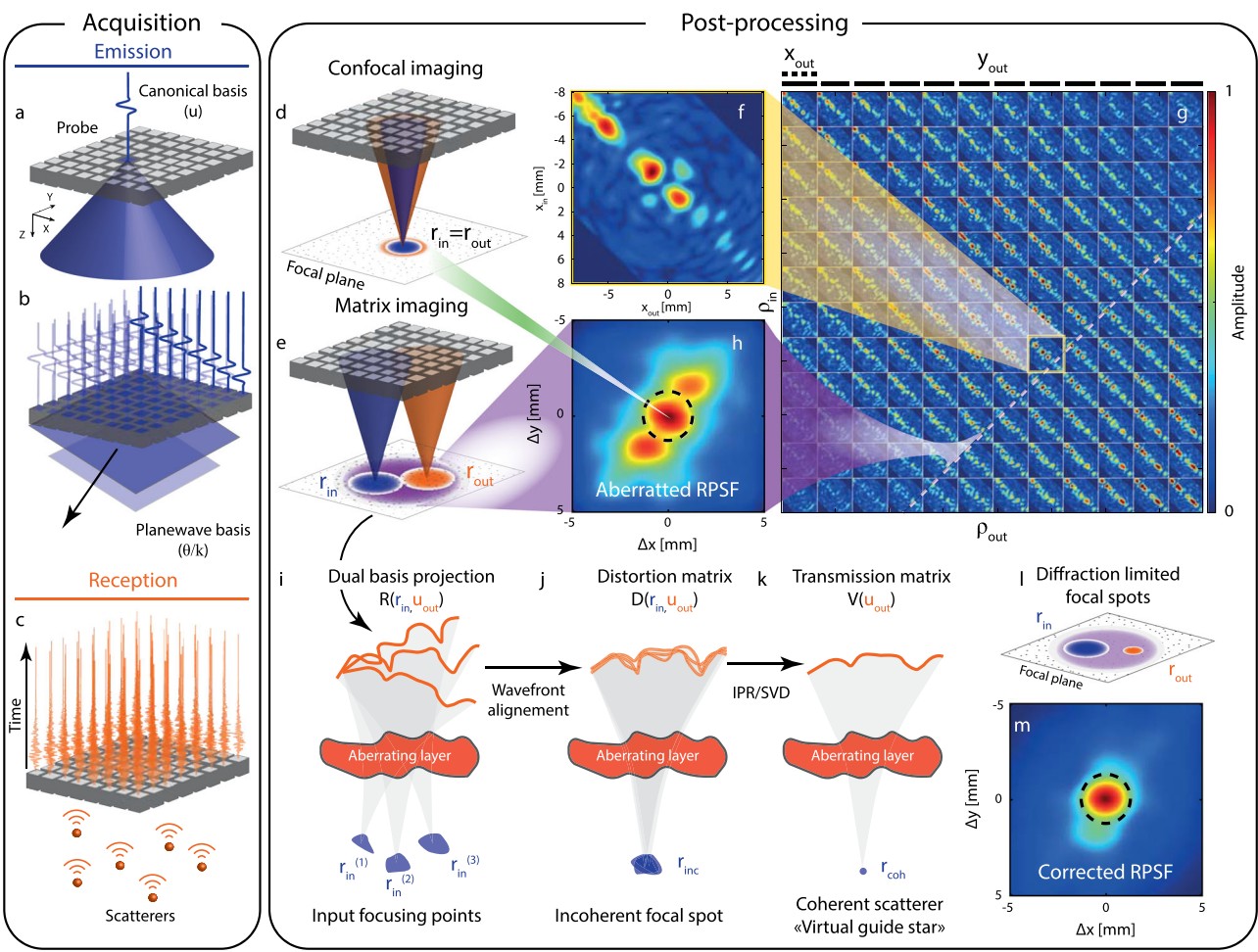

**Fig. 1 | 3D Ultrasound Matrix Imaging (UMI). a–c** The **R**-matrix can be acquired in the (**a**) transducer or (**b**) plane-wave basis in transmit and (**c**) recording the backscattered wave field on each transducer in receive. **d** Confocal imaging consists in a simultaneous focusing of waves at input and output. **e** In UMI, the input ($\mathbf{r_{in}}$) and output ($\mathbf{r_{out}}$) focusing points are decoupled. **f** $x$–cross-section of the focused **R**−matrix. **g** Four-dimensional structure of the focused **R**-matrix. **h** UMI enables a quantification of aberrations by extracting a local RPSF (displayed here in amplitude) from each antidiagonal of $\mathbf{R_{pp}}(z)$. **i** UMI then consists in a projection of the focused **R**-matrix in a correction (here transducer) basis at output. The resulting

dual **R**-matrix connects each focusing point to its reflected wave-front. **j** UMI then consists in realigning those wave-fronts to isolate their distorted component from their geometrical counterpart, thereby forming the **D**-matrix. **k** An iterative phase reversal algorithm provides an estimator of the **T**-matrix between the correction basis and the mid-point of input focusing points considered in panel (**i**). **l** The phase conjugate of the **T**-matrix provides a focusing law that improves the focusing process at output. **m** RPSF amplitude after the output UMI process. The ultrasound data shown in this figure corresponds to the pork tissue experiment at depth $z = 40$ mm.

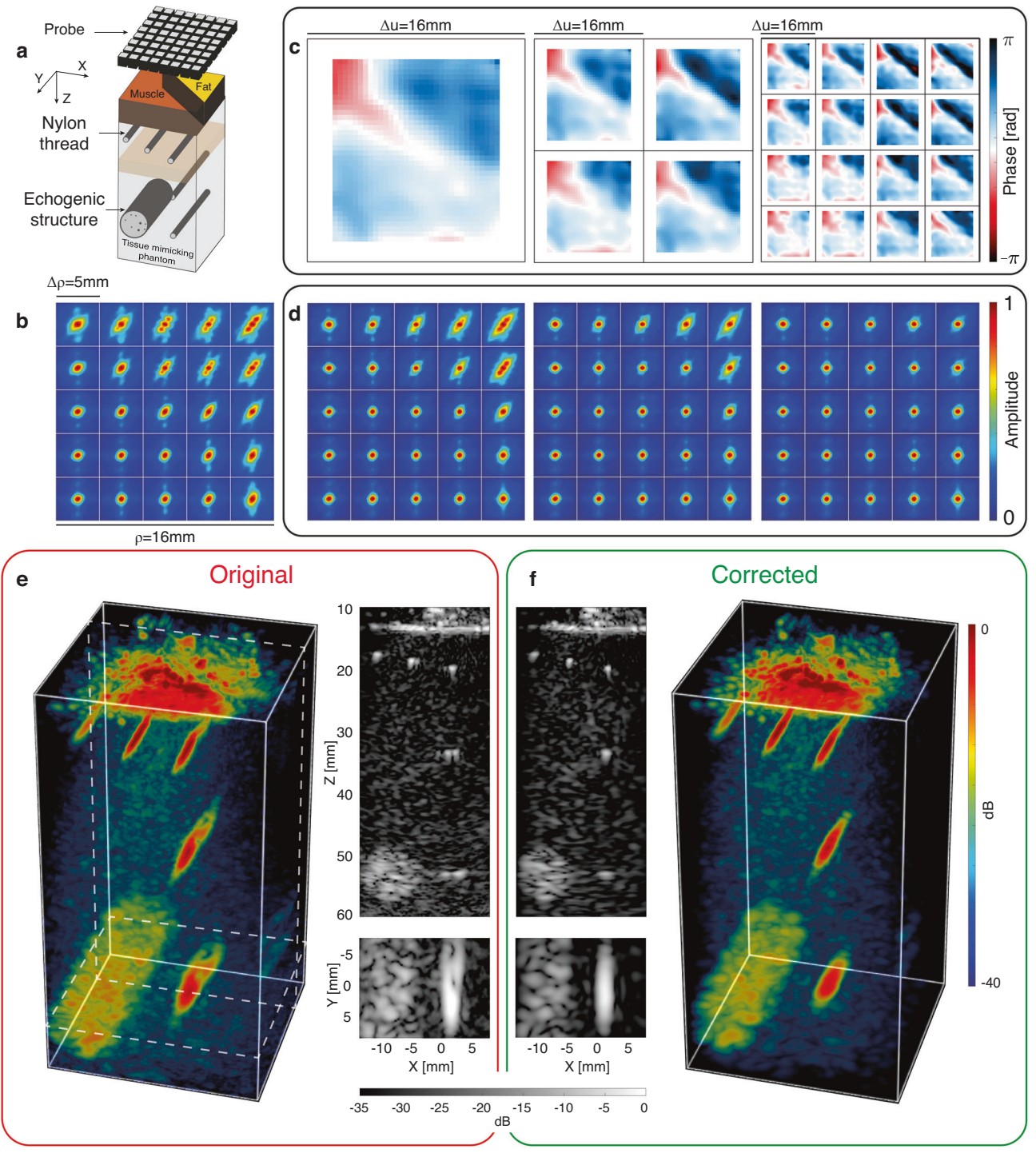

**Fig. 2 | Ultrasound Matrix Imaging of a tissue-mimicking phantom through a pork tissue. a** Schematic of the experiment. **b** Maps of original RPSFs (in amplitude) at depth $z = 29$ mm. **c** Aberration phase laws extracted at the different steps of the UMI process. **d** Corresponding RPSFs after aberration compensation at each step. **e**, **f** 3D confocal and UMI images with one longitudinal and transverse cross-section.

attenuation imposes a plane wave insonification sequence [Fig. 1b] to improve the signal-to-noise ratio. The reflection matrix $\mathbf{R}_{\theta u}$ then contains the reflected wave field $R(\boldsymbol{\theta}_{\text{in}}, \mathbf{u}_{\text{out}}, t)$ recorded by the transducers $\mathbf{u}_{\text{out}}$ [Fig. 1c] for each incident plane wave of angle $\boldsymbol{\theta}_{\text{in}}$.

Whatever the illumination sequence, the reflectivity of a medium at a given point $\mathbf{r}$ can be estimated in post-processing by a coherent compound of incident waves delayed to virtually focus on this point, and coherently summing the echoes recorded by the probe coming from that same point [Fig. 1d]. UMI basically consists in decoupling the input ($\mathbf{r}_{\text{in}}$) and output ($\mathbf{r}_{\text{out}}$) focusing points [Fig. 1e]. By applying

appropriate time delays to the transmission ($\mathbf{u}_{\text{in}}/\boldsymbol{\theta}_{\text{in}}$) and reception ($\mathbf{u}_{\text{out}}$) channels (see Methods), $\mathbf{R}_{\mathbf{uu}}(t)$ and $\mathbf{R}_{\theta u}(t)$ can be projected at each depth $z$ in a focused basis, thereby forming a broadband focused reflection matrix, $\mathbf{R}_{\rho\rho}(z) \equiv [R(\boldsymbol{\rho}_{\text{in}}, \boldsymbol{\rho}_{\text{out}}, z)]$.

Since the focal plane is bi-dimensional, each matrix $\mathbf{R}_{\rho\rho}(z)$ has a four-dimension structure: $R(\boldsymbol{\rho}_{\text{in}}, \boldsymbol{\rho}_{\text{out}}, z) = R(\{x_{\text{in}}, y_{\text{in}}\}, \{x_{\text{out}}, y_{\text{out}}\}, z)$. $\mathbf{R}_{\rho\rho}(z)$ is thus concatenated in 2D as a set of block matrices to be represented graphically [Fig. 1g]. In such a representation, every sub-matrix of $\mathbf{R}$ corresponds to the reflection matrix between lines of virtual transducers located at $y_{\text{in}}$ and $y_{\text{out}}$, whereas every element

in the given sub-matrix corresponds to a specific couple $(x_{in}, x_{out})$ [Fig. 1f]. Each coefficient $R(x_{in}, y_{in}, x_{out}, y_{out}, z)$ corresponds to the complex amplitude of the echoes coming from the point $\mathbf{r}_{out} = (x_{out}, y_{out}, z)$ in the focal plane when focusing at point $\mathbf{r}_{in} = (x_{in}, y_{in}, z)$ (or conversely, since $\mathbf{R}_{\rho\rho}(z)$ is a symmetric matrix due to spatial reciprocity).

As already shown with 2D UMI, the diagonal of $\mathbf{R}_{\rho\rho}(z)$ directly provides the transverse cross-section of the confocal ultrasound image:

$$\mathcal{I}(\boldsymbol{\rho}, z) = |R(\boldsymbol{\rho}_{in} = \boldsymbol{\rho}_{out}, z)|^2 \tag{1}$$

where $\boldsymbol{\rho} = \boldsymbol{\rho}_{in} = \boldsymbol{\rho}_{out}$ is the transverse coordinate of the confocal point. The corresponding 3D image is displayed in Fig. 2e for the pork tissue experiment. Longitudinal and transverse cross-sections illustrate the effect of the aberrations induced by the pork layer by highlighting the distortion exhibited by the image of the deepest nylon rod.

## Probing the focusing quality

We now show how to quantify aberrations in ultrasound speckle (without any guide star) by investigating the antidiagonals of $\mathbf{R}_{\rho\rho}(z)$. In the single scattering regime, the focused $\mathbf{R}$-matrix coefficients can be expressed as follows[1]:

$$R(\boldsymbol{\rho}_{out}, \boldsymbol{\rho}_{in}, z) = \int d\boldsymbol{\rho} H_{out}(\boldsymbol{\rho} - \boldsymbol{\rho}_{out}, \boldsymbol{\rho}_{out}, z) \gamma(\boldsymbol{\rho}, z) H_{in}(\boldsymbol{\rho} - \boldsymbol{\rho}_{in}, \boldsymbol{\rho}_{in}, z) \tag{2}$$

with $H_{in/out}$, the input/output point spread function (PSF); and $\gamma$ the medium reflectivity. This last equation shows that each pixel of the ultrasound image (diagonal elements of $\mathbf{R}_{\rho\rho}(z)$) results from a convolution between the sample reflectivity and an imaging PSF, which is itself a product of the input and output PSFs. The off-diagonal points in $\mathbf{R}_{\rho\rho}(z)$ can be exploited for a quantification of the focusing quality at any pixel of the ultrasound image by extracting each antidiagonal. Such an operation is mathematically equivalent to a change of variable to express the focused $\mathbf{R}$-matrix in a common midpoint basis[1] (see Supplementary Section 2):

$$R_{\mathcal{M}}(\Delta\boldsymbol{\rho}, \mathbf{r}_m) = R\left(\boldsymbol{\rho}_m - \frac{\Delta\boldsymbol{\rho}}{2}, \boldsymbol{\rho}_m + \frac{\Delta\boldsymbol{\rho}}{2}, z\right), \tag{3}$$

where the subscript $\mathcal{M}$ stands for the common midpoint basis. $\mathbf{r}_m = \{\boldsymbol{\rho}_m, z\} = \{(\boldsymbol{\rho}_{in} + \boldsymbol{\rho}_{out})/2, z\}$ is the common midpoint between the input and output focal spots, with the two separated by a distance $\Delta\boldsymbol{\rho} = \boldsymbol{\rho}_{out} - \boldsymbol{\rho}_{in}$.

In the speckle regime (random reflectivity), this quantity probes the local focusing quality as its ensemble average intensity, which we refer to as the *reflection point spread function* (RPSF), scales as an incoherent convolution between the input and output PSFs[1]:

$$RPSF(\Delta\boldsymbol{\rho}, \mathbf{r}_m) = \left\langle |R_{\mathcal{M}}(\Delta\boldsymbol{\rho}, \mathbf{r}_m)|^2 \right\rangle \propto |H_{in}|^2 \overset{\Delta\boldsymbol{\rho}}{\circledast} |H_{out}|^2 (\Delta\boldsymbol{\rho}, \mathbf{r}_m), \tag{4}$$

where $\langle \cdots \rangle$ denotes an ensemble average, which, in practice, is performed by a local spatial average (see Methods).

Figure 1h displays the mean RPSF associated with the focused $\mathbf{R}$-matrix displayed in Fig. 1g (pork tissue experiment). It clearly shows a distorted RPSF which spreads well beyond the diffraction limit [black dashed line in Fig. 1h]:

$$\delta\rho_0(z) \sim \frac{\lambda_c}{2 \sin\{\arctan[\Delta u/(2z)]\}} \tag{5}$$

with $\Delta u$ the lateral extension of the probe. The RSPF also exhibits a strong anisotropy that could not have been grasped by 2D UMI. As we

will see in the next section, this kind of aberrations can only be compensated through a 3D control of the wave field.

## Adaptive focusing by iterative phase reversal

Aberration compensation in the UMI framework is performed using the distortion matrix concept. Introduced for 2D UMI[17,28], the distortion matrix can be obtained by: (i) projecting the focused $\mathbf{R}$-matrix either at input or output in a correction basis [here the transducer basis, see Fig. 1i]; (ii) extracting wave distortions exhibited by $\mathbf{R}$ when compared to a reference matrix that would have been obtained in an ideal homogeneous medium of wave velocity $c_0$ [Fig. 1j]. The resulting distortion matrix $\mathbf{D} = [D(\mathbf{u}, \mathbf{r})]$ contains the aberrations induced when focusing on any point $\mathbf{r}$, expressed in the correction basis.

This matrix exhibits long-range correlations that can be understood in light of isoplanicity. If in a first approximation, the pork tissue layer can be considered as a phase screen aberrator, then the input and output PSFs can be considered as spatially invariant: $H_{in/out}(\boldsymbol{\rho} - \boldsymbol{\rho}_{in/out}, \mathbf{r}_{in/out}) = H(\boldsymbol{\rho} - \boldsymbol{\rho}_{in/out})$. UMI consists in exploiting those correlations to determine the transfer function $T(\mathbf{u})$ of the phase screen. In practice, this is done by considering the correlation matrix $\mathbf{C} = \mathbf{D} \times \mathbf{D}^\dagger$. The correlation between distorted wave fields enables a virtual reflector synthesized from the set of output focal spots[17] [Fig. 1k]. While, in previous works[17,19], an iterative time-reversal process (or equivalently a singular value decomposition of $\mathbf{D}$) was performed to converge towards the incident wavefront that focuses perfectly through the medium heterogeneities onto this virtual scatterer, here an iterative phase reversal algorithm is employed to build an estimator $\hat{T}(\mathbf{u})$ of the transfer function (see Methods). Supplementary Figure 3 demonstrates the superiority of this algorithm compared to SVD for 3D UMI.

Iterative phase reversal provides an estimation of aberration transmittance [Fig. 1k] whose phase conjugate is used to compensate for wave distortions (see Methods). The resulting mean RPSF is displayed in Fig. 1m. Although it shows a clear improvement compared with the initial RPSF, high-order aberrations still subsist. Because of its 3D feature, the pork tissue layer cannot be fully reduced to an aberrating phase screen in the transducer basis.

## Spatial reciprocity as a guide star

The 3D distribution of the speed-of-sound breaks the spatial invariance of input and output PSFs. Figure 2b illustrates this fact by showing a map of local RPSFs (see Methods). The RPSF is more strongly distorted below the fat layer of the pork tissue ($c_f \approx 1480 \pm 10$ m/s[36]) than below the muscle area ($c_m \approx 1560 \pm 50$ m/s). A full-field compensation of aberrations similar to adaptive focusing does not allow a fine compensation of aberrations [left panel of Fig. 2d]. Access to the transmission matrix $\mathbf{T} = [T(\mathbf{u}, \mathbf{r})]$ linking each transducer and each medium voxel is required rather than just a simple aberration transmittance $T(\mathbf{u})$.

To that aim, a local correlation matrix $\mathbf{C}(\mathbf{r}_p)$ should be considered around each point $\mathbf{r}_p$ over a sliding box $\mathcal{W}(\mathbf{r} - \mathbf{r}_p)$ (see Methods), commonly called patches, whose choice of spatial extent $w$ is subject to the following dilemma: On the one hand, the spatial window should be as small as possible to grasp the rapid variations of the PSFs across the field of view; on the other hand, these areas should be large enough to encompass a sufficient number of independent realizations of disorder[16,19]. The bias made on our $\mathbf{T}$-matrix estimator actually scales as (see Supplementary Section 6):

$$|\delta T(\mathbf{u}, \mathbf{r}_p)|^2 \sim \frac{1}{\mathcal{C}^2 N_{\mathcal{W}}}. \tag{6}$$

$\mathcal{C}$ is the so-called coherence factor that is a direct indicator of the focusing quality[8] but that also depends on the multiple scattering rate and noise background[28]. $N_{\mathcal{W}}$ is the number of diffraction-limited resolution cells in each spatial window.

The validity of the **T**-matrix estimator in a region $\mathcal{W}_1$ [Fig. 3c] is investigated by examining the corrected RPSF in a neighbor region $\mathcal{W}_2$ (yellow box). $\mathcal{W}_1$ and $\mathcal{W}_2$ are sufficiently close to assume, in a first approximation, that they belong to the same isoplanatic patch. If the box is too small [left panels of Fig. 3d], our estimator has not converged yet and the correction is not valid, as shown by the degraded quality of the RPSF in $\mathcal{W}_2$ [left panels of Fig. 3h] compared to its initial value [Fig. 3g]. With sufficient spatial averaging [third panel of Fig. 3d], a valid aberration law can be extracted, as shown by a corrected RPSF now close to be only diffraction-limited [third panel of Fig. 3h].

The question that now arises is how we can, in practice, know if the convergence of $\hat{\mathbf{T}}$ is fulfilled without any a priori knowledge on **T**. An answer can be found by comparing the estimated input and output aberration phase laws, $\hat{T}_{in}(\mathbf{u},\mathbf{r}_p)$ and $\hat{T}_{out}(\mathbf{u},\mathbf{r}_p)$, at a given point $\mathbf{r}_p$ as shown in Fig. 3e, f. Spatial reciprocity implies that $\hat{T}_{in}$ and $\hat{T}_{out}$ shall be equal when the convergence of the estimator is reached [third panel of Fig. 3e, f]. Their normalized scalar product, $P_{in/out} = N_u^{-1}|\hat{\mathbf{T}}_{in}\hat{\mathbf{T}}_{out}^{\dagger}|$, can thus be used to probe the error made on the aberration phase law $|\delta T|^2$. Both quantities are actually related as follows (see Supplementary Section 7):

$$|\delta T|^2 \simeq 1 - P_{in/out}. \tag{7}$$

The normalized scalar product $P_{in/out}$ is displayed as a function of $w$ and shows the convergence of the IPR process [Fig. 3a]. For a sufficiently large box [third panel of Fig. 3d], $\hat{\mathbf{T}}$ is supposed to have converged towards **T** when $\hat{\mathbf{T}}_{in}$ and $\hat{\mathbf{T}}_{out}$ are almost equal [third panel of Fig. 3e,f], while, for a small box [left panels of Fig. 3d], a large discrepancy can be found between them. In the following, the parameter $P_{in/out}$ will thus be used as a guide star for monitoring the convergence of the UMI process.

The scaling law of Eq. (6) with respect to $N_{\mathcal{W}}$ is checked in Fig. 3b. The inverse scaling of the bias with $N_{\mathcal{W}}$ shows the advantage of 3D UMI over 2D UMI, since $N_{\mathcal{W}} \sim w^d$, with $d$ the imaging dimension. This superiority is evident in Fig. 3a, which shows a faster convergence with 3D boxes (green curve) than with 2D patches (orange curve). For a given precision, 3D UMI thus provides a better spatial resolution for our **T**-matrix estimator as shown by right panels of Fig. 3f, where much

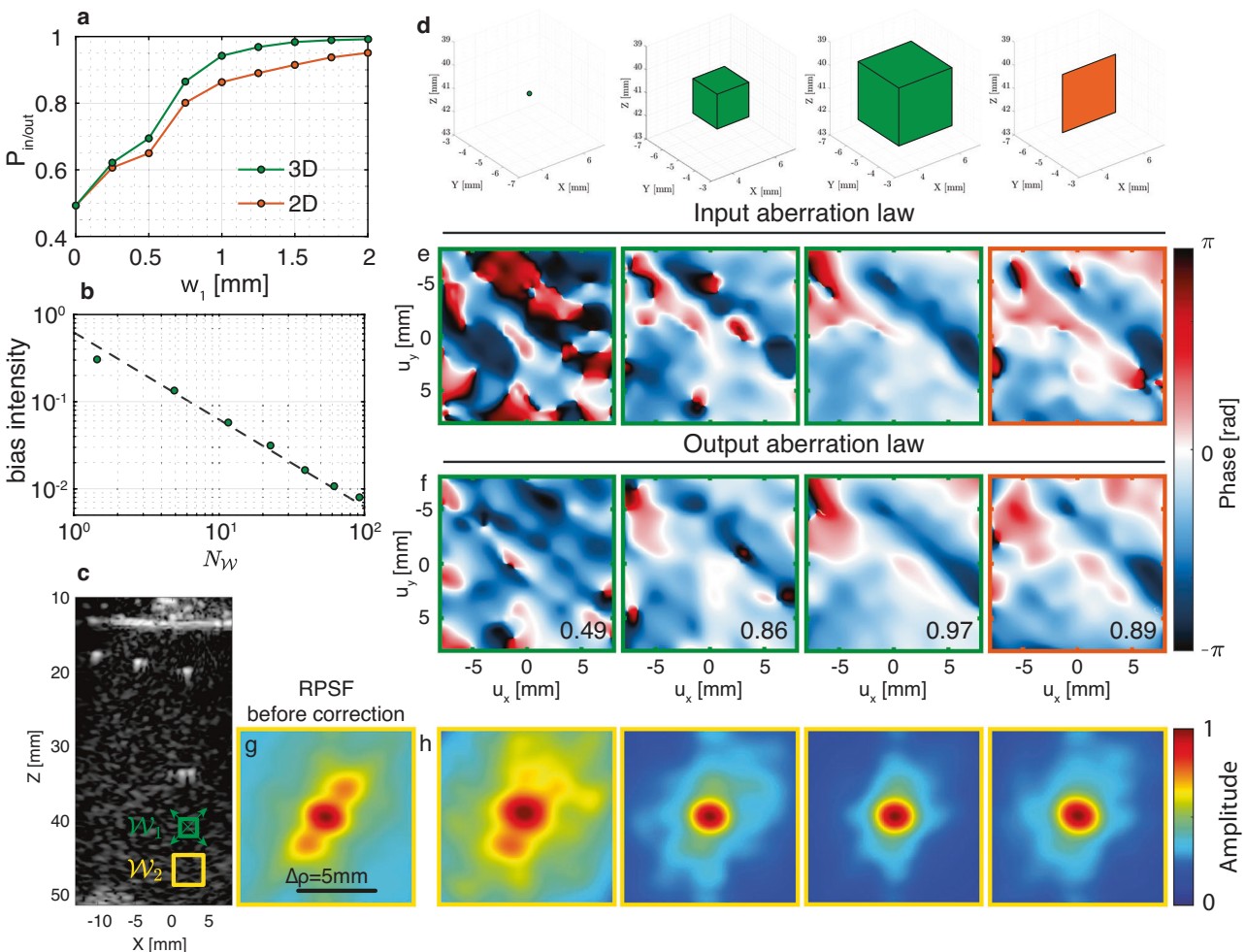

**Fig. 3 | Convergence of the UMI process towards the T-matrix. a** Normalized scalar product $P_{in/out}$ extracted at a point $\mathbf{r}_1$ (**c**) as a function of the size $w_1$ of the considered spatial window $\mathcal{W}_1$ for 2D (orange) and 3D (green) imaging. **b** Corresponding bias intensity estimator, $|\delta T|^2 = 1 - P_{in/out}$, as a function of the number of resolution cells $N_{\mathcal{W}}$ contained in the window $\mathcal{W}_1$. The plot is in log-log scale and the theoretical power law (Eq. (6)) is shown with a dashed black line for comparison. **c** Cross-section of the confocal volume showing the location of $\mathcal{W}_1$ in green and $\mathcal{W}_2$ in yellow. The green box $\mathcal{W}_1$, centered around the point $\mathbf{r}_1 = (5, -5, 41)$ mm, denotes the region where the $\hat{\mathbf{T}}$-matrix is extracted, while the

yellow box $\mathcal{W}_2$, of fixed size $w_2 = 2$ mm and centered around the point $\mathbf{r}_2 = (5, -5, 45)$ mm, is the area where the effect of aberration correction is investigated by means of the RPSF. **d** Spatial windows $\mathcal{W}_1$ considered for the calculation of **C**($\mathbf{r}_1$). From left to right: Boxes of dimension $w = 0$ mm, $w = 0.75$ mm, $w = 1.25$ mm, rectangle of dimension $w = 1.25$ mm. **e**, **f** Corresponding input $\hat{\mathbf{T}}_{in}$ and output $\hat{\mathbf{T}}_{out}$ aberration laws, respectively. The scalar product $P_{in/out}$ is displayed in each subpanel of (**f**). **g**, **h** RPSF associated with the yellow box $\mathcal{W}_2$ (**g**) before correction and (**h**) after correction using the corresponding $\hat{\mathbf{T}}$-matrices displayed in panels (**e**) and (**f**).

better agreement between $\hat{\mathbf{T}}_{\text{in}}$ and $\hat{\mathbf{T}}_{\text{out}}$ is observed for a 3D box [third panel of Fig. 3d] than for a 2D patch [right panel of Fig. 3d] of same dimension $w$.

## Multi-scale compensation of wave distortions

The scaling of the bias intensity $|\delta T|^2$ with the coherence factor $\mathcal{C}$ has not been discussed yet. This dependence is however crucial since it indicates that a gradual compensation of aberrations shall be favored rather than a direct partition of the field of view into small boxes[22] [see Supplementary Fig. 4]. An optimal UMI process should proceed as follows: first, compensate for input and output wave distortions at a large scale to increase the coherence factor $\mathcal{C}$; then, decrease the spatial window $\mathcal{W}$ and improve the resolution of the **T**-matrix estimator. The whole process can be iterated, leading to a multi-scale compensation of wave distortions (see Methods). As explained above, the convergence of the process is monitored using spatial reciprocity ($P_{\text{in/out}} > 0.9$).

The result of 3D UMI is displayed in Fig. 2. It shows the evolution of the **T**-matrix at each step [Fig. 2c] and the corresponding local RPSFs [Fig. 2d]. In the most aberrated area (i.e. under the fat), the phase fluctuations of the aberration law corresponds to a time delay spread of 56 ns (rms). This value is comparable with past measurements through the human abdominal wall[37]. The pork tissue layer thus induces a level of aberrations typical of standard ultrasound diagnosis. The comparison with the initial and full-field maps of RPSF highlights the benefit of a local compensation via the **T**-matrix, with a diffraction-limited resolution reached everywhere. The local aberration phase laws exhibited by $\hat{\mathbf{T}}$ perfectly match with the distribution of muscle and fat in the pork tissue layer. The comparison of the final 3D image [Fig. 2f] and its cross-sections with their initial counterparts [Fig. 2e] show the success of the UMI process, in particular for the deepest nylon rod, which has retrieved its straight shape. The local RPSF on the top right of Fig. 2 shows a contrast improvement by 4.2 dB and resolution enhancement by a factor 2 [see Methods and Supplementary Fig. 5].

## Overcoming multiple scattering for trans-cranial imaging

The same UMI process is now applied to the ultrasound data collected on the head phantom [Fig. 4a]. The parameters of the multi-scale analysis are provided in the Methods section [see also Supplementary Fig. 6]. The first difference with the pork tissue experiment lies in our choice of correction basis. Given the multi-layer configuration in this experiment, the **D**-matrix is investigated in the plane wave basis[17].

The second difference is that our spatial reciprocity criterion $P_{\text{in/out}}$ is very low [see the blue box plot in Fig. 4e]. This is the manifestation of a bad convergence of our **T**-matrix estimator. The incoherent background exhibited by the original PSFs [Fig. 5c] drastically affects the coherence factor $\mathcal{C}$[28], which, in return, gives rise to a strong bias on the **T**-matrix estimator (Eq. (6)). The incoherent background is due to multiple scattering events in the skull and electronic noise, whose relative weight can be estimated by investigating the spatial reciprocity symmetry of the **R**-matrix (see Methods). Figure 5b shows the depth evolution of the single and multiple scattering contributions, as well as electronic noise. While single scattering dominates at shallow depths ($z < 20$ mm), multiple scattering quickly reaches 35% and remains relatively constant until electronic noise increases, so that the three contributions are almost equal at depths of 75 mm.

Beyond the depth evolution, 3D imaging even allows the study of multiple scattering in the transverse plane, as shown in Fig. 5a. Two areas are examined, marked with black boxes, corresponding to the RPSFs shown in [Fig. 5c] ($z = 32$ mm). In the center, the RPSFs exhibits a low background due to the presence of a spherical target, resulting in a single scattering rate of 90%. The second box on the right, however, is characterized by a much higher background, leading to a multiple-to-single scattering ratio slightly larger than one. This high level of multiple scattering highlights the difficult task of trans-cranial imaging with ultrasonic waves.

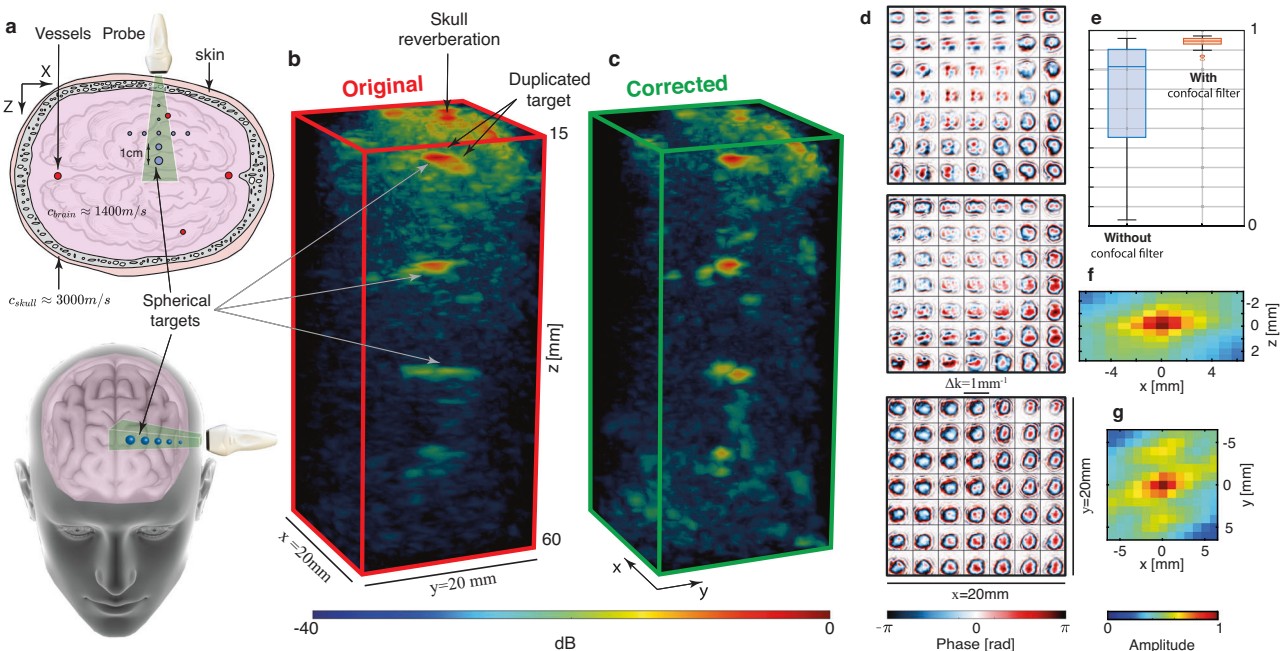

**Fig. 4 | Ultrasound Matrix Imaging of the head phantom. a** Top and oblique views of the experimental configuration. Image credits: Harryarts and kjpargeter on Freepik. **b**, **c** Original and UMI images, respectively. **d** Aberration laws at 3 different depths. From top to bottom: $z = 20$ mm, $z = 32$ mm, $z = 60$ mm. **e** Reciprocity criterion $P_{\text{in/out}}$ with or without the use of a confocal filter: Each box chart displays the median, lower and upper quartiles, and the minimum and maximum values. **f**, **g**. Correlation function of the $\hat{\mathbf{T}}$-matrix in the $(x, z)$-plane (**f**) and $(x, y)$-plane (**g**), respectively. We attribute the sidelobes along the $y$-axis (**g**) to the inactive rows separating each block of 256 elements of the matrix array.

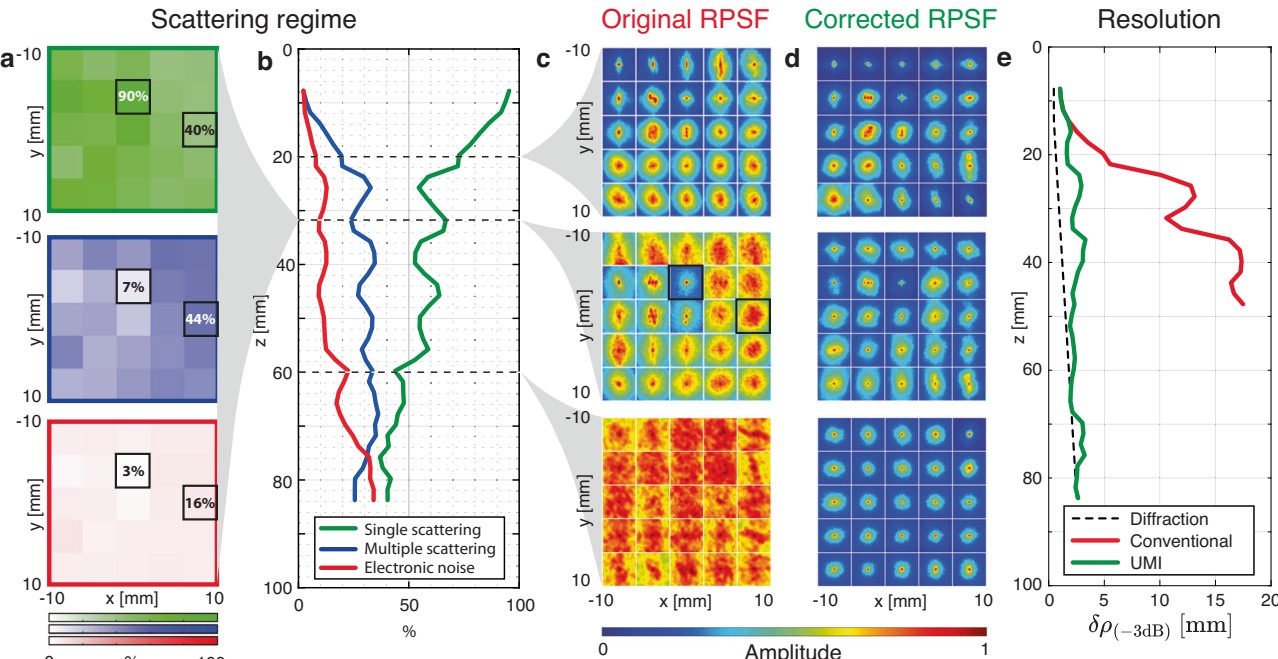

**Fig. 5 | Aberrations and multiple scattering quantification in the head phantom. a** Single scattering (green), multiple scattering (blue) and noise (red) rate at $z = 32$ mm. **b** Single scattering, multiple scattering, and noise rates as a function of depth. **c**, **d** Maps of local RPSFs (in amplitude) before and after correction, respectively, at three different depths (From top to bottom: $z = 20$ mm, 32 mm and 60 mm. Black boxes in panel (**a**) and (**c**) corresponds to the same area. **e** Resolution $\delta\rho_{(-3dB)}$ as a function of depth. Initial resolution (red line) and its value after UMI (green line) are compared with the ideal (diffraction-limited) resolution (Eq. (5)).

In order to overcome these detrimental effects, an adaptive confocal filter can be applied to the focused **R**-matrix[19].

$$R'(\boldsymbol{\rho}_{\text{in}}, \boldsymbol{\rho}_{\text{out}}, z) = R(\boldsymbol{\rho}_{\text{in}}, \boldsymbol{\rho}_{\text{out}}, z)\exp\left(-\frac{|\boldsymbol{\rho}_{\text{out}} - \boldsymbol{\rho}_{\text{in}}|^2}{2l_c(z)^2}\right) \quad (8)$$

This filter has a Gaussian shape, with a width $l_c(z)$ that scales as $3\delta\rho_0(z)$[19]. The application of a confocal filter drastically improves the correlation between input and output aberration phase laws [see Fig. 4e and Supplementary Fig. 7], proof that a satisfying convergence towards the **T**-matrix is obtained.

Figure 4d shows the **T**-matrix obtained at different depths in the brain phantom. Its spatial correlation function displayed in Fig. 4f, g provides an estimation of the isoplanatic patch size: 5 mm in the transverse direction [Fig. 4g] and 2 mm in depth [Fig. 4f]. This rapid variation of the aberration phase law across the field of view confirms *a posteriori* the necessity of a local compensation of aberrations induced by the skull. It also confirms the importance of 3D UMI with a fully sampled 2D array, as previous work recommended that the array pitch should be no more than 50% of the aberrator correlation length to properly sample the corresponding adapted focusing law[38].

The phase conjugate of the **T**-matrix at input and output enables a fine compensation of aberrations. A set of corrected RPSFs are shown in Fig. 5d. The comparison with their initial values demonstrates the success of 3D UMI: a diffraction-limited resolution is obtained almost everywhere [Fig. 5e], whether it be in ultrasound speckle or in the neighborhood of bright targets, at shallow or high depths, which proves the versatility of UMI.

The performance of 3D UMI is also striking when comparing the three-dimensional image of the head phantom before and after UMI [Fig. 4b, c, respectively]. The different targets were initially strongly distorted by the skull, and are now nicely resolved with UMI. In particular, the first target, located at $z = 19$ mm and originally duplicated, has recovered its true shape. In addition, two targets laterally spaced by 10 mm are observed at 42 mm depth, as expected [Fig. 4a]. The image of the target observed at 54 mm depth is also drastically improved in terms of contrast and resolution but is not found at the expected transverse position. One potential explanation is the size of this target (2 mm diameter) larger than the resolution cell. The guide star is thus far from being point-like, which can induce an uncertainty on the absolute transverse position of the target in the corrected image.

Finally, an isolated target can be leveraged to highlight the gain in contrast provided by 3D UMI with respect to its 2D counterpart. To that aim, a linear 1D array is emulated from the same raw data by collimating the incident beam in the *y*-direction [Fig. 6]. The ultrasound image is displayed before and after UMI in Fig. 6b, c, respectively. The radial average of the corresponding focal spots is displayed in Fig. 6d. Even though 2D UMI enables a diffraction-limited resolution, the contrast gain $G$ is quite moderate ($G_{2D} \sim 8$dB) as it scales with the number $N$ of coherence grains exhibited by the 1D aberration phase law [Fig. 6a]: $N_{2D} \sim 6.2$. On the contrary, as expected, 3D UMI provides a strong enhancement of the target echo [see the comparison between Fig. 6e–g]: $G_{3D} \sim 18$ dB. The 2D aberration phase law actually provides a much larger number of spatial degrees of freedom than its 1D counterpart: $N_{3D} \sim 63$. The gain in contrast is accompanied by a drastic improvement of the transverse resolution [>8× for $z > 40$ mm in Fig. 5e]. Figure 6 demonstrates the necessity of a 2D ultrasonic probe for trans-cranial imaging. Indeed, the complexity of wave propagation in the skull can only be harnessed with a 3D control of the incident and reflected wave fields.

## Discussion

In this experimental proof-of-concept, we demonstrated the capacity of 3D UMI to correct strong aberrations such as those encountered in trans-cranial imaging. This work is not only a 3D extension of previous studies[17,28] since several crucial elements have been introduced to make UMI more robust.

First, the proposed iterative phase reversal algorithm outperforms the SVD for local compensation of aberrations because it can

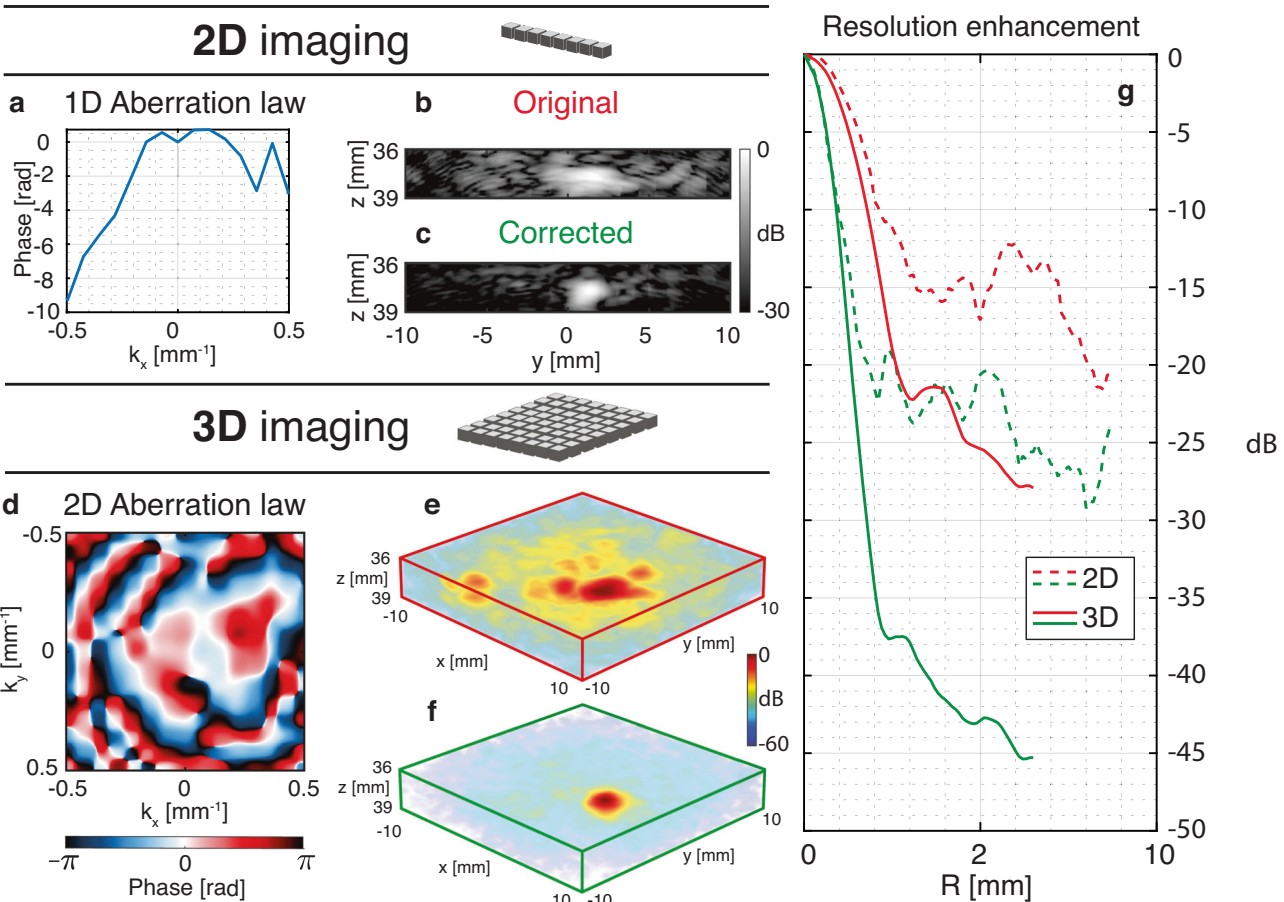

**Fig. 6 | 2D *versus* 3D matrix imaging in a head phantom. a** Aberration law extracted with 2D UMI for a target located at $z = 38$ mm. **b**, **c** Original and corrected images of the same target with 2D UMI, respectively. **d** Aberration law extracted with 3D UMI. **e**, **f** Original and corrected images of the same target with 3D UMI, respectively. **g** Imaging PSF before (red) and after (green) 2D (dotted line) and 3D (solid line) UMI. The depth range considered in each panel corresponds to the echo of the target located at $z = 38$ mm.

evaluate the aberration law on a larger angular support [see Supplementary Fig. 3], resulting in a sharper compensation of aberrations. Second, the bias of our **T**-matrix estimator has been expressed analytically (Eq. (6)) as a function of the coherence factor that grasps the detrimental effects of the virtual guide star blurring induced by aberrations, multiple scattering and noise. This led us to define a general strategy for UMI with: (*i*) a multi-scale compensation of wave distortions to gradually reduce the blurring of the virtual guide star and tackle high-order aberrations associated with small isoplanatic lengths; (*ii*) the application of an adaptive confocal filter to cope with multiple scattering and noise; (*iii*) a fine monitoring of the convergence of our estimator by means of spatial reciprocity. The latter is a real asset, as it provides an objective criterion to check the physical significance of the extracted aberration laws and optimize the resolution of our **T**-matrix estimator.

Although the results presented in this paper are striking, they were obtained in vitro, and some challenges remain for in vivo brain imaging. Until now, UMI has only been applied to a static medium, while biological tissues are usually moving, especially in the case of vascular imaging, where blood flow makes the reflectivity vary quickly over time. A lot of 3D imaging modes are indeed designed to image blood flow, such as transcranial Doppler imaging[39] or ULM[40,41]. These methods are strongly sensitive to aberrations[42,43] and their coupling with matrix imaging would be rewarding to increase the signal-to-noise ratio and improve the image resolution, not only in the vicinity of bright reflectors[44] but also in ultrasound speckle.

However, due to spatial aliasing, the number of illuminations required for UMI scales with the number of resolution cells covered by the RPSF [see Supplementary Fig. 8]. Because the aberration level through the skull is important, the illumination basis should thus be fully sampled. It limits 3D transcranial UMI to a compounded frame rate of only a few hertz, which is much too slow for ultrafast imaging[45]. Moreover, a reduced number of illuminations breaks the symmetry of the reflection matrix. It would therefore also affect the accuracy of our monitoring parameter based on spatial reciprocity.

Soft tissues usually exhibit much slower movement, and provide signals several dB higher than blood. Ultrasound imaging of tissues is generally discarded for the brain because of the strong level of aberrations and reverberations. Interestingly, UMI can open a new route towards quantitative brain imaging since a matrix framework can also enable the mapping of physical parameters such as the speed-of-sound[1,46–48], attenuation and scattering coefficients[49,50], or fiber anisotropy[51,52]. Those various observables can be extremely enlightening for the characterization of cerebral tissues.

Alternatively, a solution to directly implement 3D UMI in vivo for ultrafast imaging, would be to design an imaging sequence in which the fully sampled **R**-matrix is acquired prior to the ultrafast acquisition itself, where the illumination basis can be drastically downsampled. The $\hat{\mathbf{T}}$-matrix obtained from **R** could then be used to correct the ultrafast images in post-processing.

Interestingly, if an ultrafast 3D UMI acquisition is possible (in cases with less aberrations, or at shallow depths), the quickly decorrelating

speckle observed in blood flow can be an opportunity since it provides a large number of speckle realizations in a given voxel. A high resolution **T**-matrix could thus be, in principle, extracted without spatial averaging and relying on any isoplanatic assumption[53,54].

So far, one limit of UMI concerns the strong aberration regime in which extreme time delay fluctuations can occur. Indeed, our approach relies on a broadband focused reflection matrix that consists of a coherent time gating of singly scattered echoes. If time delay fluctuations are larger than the time resolution $\delta t$ of our measurement, the angular components of each echo will not necessarily emerge in the same time gate and aberration compensation will be imperfect.

Beyond strong aberrations, another issue for transcranial imaging arises from multiple reflections caused by the skull. While such reverberations are not observed in the pork tissue experiment, their detrimental effects are much greater in a transcranial experiment because of the large impedance mismatch between the skull and brain tissues. In this work, such artefacts are not corrected and they drastically pollute the image at shallow depths ($z < 20$ mm).

To cope with those issues, a polychromatic approach to matrix imaging is required. Indeed, the aberration compensation scheme proposed in this paper is equivalent to a simple application of time delays on each transmit and receive channel. On the contrary, a full compensation of reverberation requires the tailoring of a complex spatio-temporal adaptive (or even inverse) filter. To that aim, 3D UMI provides an adequate framework to exploit, at best, all the spatio-temporal degrees of freedom provided by a high-dimension array of broadband transducers.

To conclude, 3D UMI is general and can be applied to any insonification sequence (plane wave or virtual source illumination) or array configuration (random or periodic, sparse or dense). Matrix imaging can be also extended to any field of wave physics for which a multi-element technology is available: optical imaging[20–22], seismic imaging[25,26] and also radar[55]. All the conclusions raised in that paper can be extended to each of these fields. The matrix formalism is thus a powerful tool for the big data revolution coming in wave imaging.

## Methods
### Description of the pork tissue experiment
The first sample under investigation is a tissue-mimicking phantom (speed of sound: $c_0 = 1540$ m/s) composed of random distribution of unresolved scatterers which generate ultrasonic speckle characteristic of human tissue [Fig. 2a]. The system also contains nylon filaments placed at regular intervals, with a point-like cross-section, and, at a depth of 40 mm, a 10 mm-diameter hyperechoic cylinder, containing a higher density of unresolved scatterers. A 12-mm thick pork tissue layer is placed on top of the phantom. It is immersed in water to ensure its acoustical contact with the probe and the phantom. Since the pork layer contains a part of muscle tissue ($c_m \sim 1560$ m/s) and a part of fat tissue ($c_f \sim 1480$ m/s), it acts as an aberrating layer. This experiment mimics the situation of abdominal in vivo imaging, in which layers of fat and muscle tissues generate strong aberration and scattering at shallow depths.

The acquisition of the reflection matrix is performed using a 2D matrix array of transducers (Vermon) whose characteristics are provided in Table 1. The electronic hardware used to drive the probe was developed by Supersonic Imagine (member of Hologic group) in the context of a collaboration agreement with the Langevin Institute.

The reflection matrix is acquired by recording the impulse response between each transducer of the probe using IQ modulation with a sampling frequency $f_s = 6$ MHz. To that aim, each transducer $\mathbf{u}_{in}$ emits successively a sinusoidal burst of three half periods at the central frequency $f_c$. For each excitation $\mathbf{u}_{in}$, the back-scattered wave field is recorded by all probe elements $\mathbf{u}_{out}$ over a time length $\Delta t = 139 \mu s$. This set of impulse responses is stored in the canonical reflection matrix $\mathbf{R_{uu}}(t) = [R(\mathbf{u}_{in}, \mathbf{u}_{out}, t)]$.

## Table 1 | Matrix array datasheet

| Number of transducers | 32 × 32 = 1024 (with 6 dead elements) |
|---|---|
| Geometry ($y$-axis) | 3 inactive rows between each block of 256 elements |
| Pitch | $\delta u = 0.5$ mm ($\approx \lambda$ at $c = 1540$ m/s) |
| Aperture | $\Delta\mathbf{u} = \begin{pmatrix} \Delta u_x \\ \Delta u_y \end{pmatrix} = \begin{pmatrix} 16 \text{ mm} \\ 17.5 \text{ mm} \end{pmatrix}$ |
| Central frequency | $f_c = 3$ MHz |
| Bandwidth (at −6 dB) | 80% → $\Delta f = [1.8–4.2]$ MHz |
| Transducer directivity | $\theta_{max} = 28°$ at $c = 1400$ m/s |

## Table 2 | Head phantom characteristics

| | Speed-of-sound [m/s] | Density [g/cm³] | Attenuation at 2.25 MHz [dB/cm] |
|---|---|---|---|
| Cortical bone | 3000 ± 30 | 2.31 | 6.4 ± 0.3 |
| Trabecular bone | 2800 ± 50 | 2.03 | 21 ± 2 |
| Brain tissue | 1400 ± 10 | 0.99 | 1.0 ± 0.2 |
| Skin tissue | 1400 ± 10 | 1.01 | 1.7 ± 0.2 |

### Description of the head phantom experiment
In this second experiment, the same probe [Table 1] is placed slightly above the temporal window of a mimicking head phantom, whose characteristics are described in Table 2. To investigate the performance of UMI in terms of resolution and contrast, the manufacturer (True Phantom Solutions) was asked to place small spherical targets made of bone-mimicking material inside the brain. They are arranged crosswise, evenly spaced in the 3 directions with a distance of 1 cm between two consecutive targets, and their diameter increases with depth: 0.2, 0.5, 1, 2, 3 mm [Fig. 4a]. Skull thickness is of ~6 mm on average at the position where the probe is placed and the first spherical target is located at $z \approx 20$ mm depth, while the center of the cross is at $z \approx 40$ mm depth. The transverse size of the head is ~14 cm.

To improve the signal-to-noise ratio, the **R**-matrix is here acquired using a set of plane waves[56]. For each plane wave of angles of incidence $\boldsymbol{\theta}_{in} = (\theta_x, \theta_y)$, the time-dependent reflected wave field $R(\boldsymbol{\theta}_{in}, \mathbf{u}_{out}, t)$ is recorded by each transducer $\mathbf{u}_{out}$. This set of wave fields forms a reflection matrix acquired in the plane wave basis, $\mathbf{R}_{\theta u}(t) = [R(\boldsymbol{\theta}_{in}, \mathbf{u}_{out}, t)]$. Since the transducer and plane wave bases are related by a simple Fourier transform at the central frequency, the array pitch $\delta u$ and probe size $\Delta u$ dictate the angular pitch $\delta\theta$ and maximum angle $\theta_{max}$ necessary to acquire a full reflection matrix in the plane wave basis such that: $\theta_{max} = \arcsin[\lambda_c/(2\delta u)] \approx 28°$; $\delta\theta = \arcsin[\lambda_c/(2\Delta u_y)] \approx 0.8°$, with $\lambda_c = c_0/f_c$ the central wavelength and $c_0 = 1400$ m/s the speed-of-sound in the brain phantom. A set of 1225 plane waves are thus generated by applying appropriate time delays $\Delta\tau(\boldsymbol{\theta}_{in}, \mathbf{u}_{in})$ to each transducer $\mathbf{u}_{in} = (u_x, u_y)$ of the probe:

$$\Delta\tau(\boldsymbol{\theta}_{in}, \mathbf{u}_{in}) = [u_x \sin\theta_x + u_y \sin\theta_y]/c_0. \quad (9)$$

### Focused beamforming of the reflection matrix
The focused **R**-matrix, $\mathbf{R}_{\rho\rho}(z) = [R(\boldsymbol{\rho}_{in}, \boldsymbol{\rho}_{out}, z)]$, is built in the time domain via a conventional delay-and-sum beamforming scheme that consists in applying appropriate time-delays in order to focus at different points at input $\mathbf{r}_{in} = (\boldsymbol{\rho}_{in}, z) = (\{x_{in}, y_{in}\}, z)$ and output $\mathbf{r}_{out} = (\boldsymbol{\rho}_{out}, z) = (\{x_{out}, y_{out}\}, z)$:

$$R(\boldsymbol{\rho}_{in}, \boldsymbol{\rho}_{out}, z) = \sum_{\mathbf{i}_{in}} \sum_{\mathbf{u}_{out}} A(\{\mathbf{i}_{in}, \mathbf{r}_{in}\}, \{\mathbf{u}_{out}, \mathbf{r}_{out}\}) R(\mathbf{i}_{in}, \mathbf{u}_{out}, \tau(\mathbf{i}_{in}, \mathbf{r}_{in})$$
$$+ \tau(\mathbf{u}_{out}, \mathbf{r}_{out})) \quad (10)$$

**Table 3 | Parameters of UMI in both experiments**

|  | Pork tissue | | | Head phantom | | | | | |
|---|---|---|---|---|---|---|---|---|---|
| Correction step | 1° | 2° | 3° | 1° | 2° | 3° | 4° | 5° | 6° |
| Number of transverse patches | 1×1 | 2×2 | 4×4 | 1×1 | 2×2 | 3×3 | 4×4 | 5×5 | 6×6 |
| $w_\rho = (w_x, w_y)$ [mm] | 16 | 12 | 8 | 20 | 15 | 13.3 | 10 | 8 | 6.6 |
| $w_z$ [mm] | 3 | 3 | 3 | 5.5 | 5.5 | 5.5 | 5.5 | 5.5 | 5.5 |

where $\mathbf{i} = \mathbf{u}$ or $\boldsymbol{\theta}$ accounts for the illumination basis. $A$ is an apodization factor that limit the extent of the synthetic aperture at emission and reception. This synthetic aperture is dictated by the transducers' directivity $\theta_{max} \sim 28°$ [57].

In the transducer basis, the time-of-flights, $\tau(\mathbf{u}, \mathbf{r})$, writes:

$$\tau(\mathbf{u},\mathbf{r}) = \frac{|\mathbf{u} - \mathbf{r}|}{c_0} = \frac{\sqrt{(x - u_x)^2 + (y - u_y)^2 + z^2}}{c_0}. \qquad (11)$$

In the plane wave basis, $\tau(\boldsymbol{\theta}, \mathbf{r})$ is given by

$$\tau(\boldsymbol{\theta},\mathbf{r}) = \left[ x \sin\theta_x + y \sin\theta_y + z\sqrt{1 - \sin^2\theta_x - \sin^2\theta_y} \right] / c_0. \qquad (12)$$

### Local average of the reflection point spread function

To probe the local RPSF, the field of view is divided into spatial regions $\mathcal{W}(\mathbf{r}_m - \mathbf{r}_p)$, defined by their center $\mathbf{r}_p$ and their extent $\mathbf{w} = (w_\rho, w_z)$, where $w_\rho$ and $w_z$ denote the lateral and axial extent, respectively. A local average of the back-scattered intensity can then be performed in each region:

$$RPSF(\Delta\boldsymbol{\rho}, \mathbf{r}_p) = \left\langle |R_{\mathcal{M}}(\Delta\boldsymbol{\rho}, \mathbf{r}_m)|^2 \mathcal{W}(\mathbf{r}_m - \mathbf{r}_p) \right\rangle_{\mathbf{r}_m} \qquad (13)$$

where the symbol $\langle \cdots \rangle$ denotes here a spatial average over the variable in the subscript. $\mathcal{W}(\mathbf{r}_m - \mathbf{r}_p) = 1$ for $|\boldsymbol{\rho}_m - \boldsymbol{\rho}_p| < w_\rho/2$ and $|z_m - z_p| < w_z/2$, and zero otherwise. The dimensions of $\mathcal{W}$ used for [Fig. 2b, d] are: $\mathbf{w} = (w_\rho, w_z) = (3.2, 3)$ mm. The dimensions of $\mathcal{W}$ to obtain [Fig. 5c, d] are: $\mathbf{w} = (w_\rho, w_z) = (4, 5.5)$ mm.

### Distortion matrix in 3D UMI

The first step consists in projecting the focused $\mathbf{R}$-matrix $\mathbf{R}_{\rho\rho}(z)$ [Fig. 1e] onto a dual basis $\mathbf{c}$ at output [Fig. 1i]:

$$\mathbf{R}_{\rho c}(z) = \mathbf{R}_{\rho\rho}(z) \times \mathbf{G}_{\rho c}(z) \qquad (14)$$

where the symbol $\times$ stands for the matrix product. $\mathbf{G}_{\rho c}(z)$ is the propagation matrix predicted by the homogeneous propagation model between the focused basis ($\rho$) and the correction basis ($\mathbf{c}$) at each depth $z$. $\mathbf{c}$ can be either the plane wave, the transducer, or any other correction basis suitable for a particular experiment[23,58,59].

In the transducer basis ($\mathbf{c} = \mathbf{u}$), the coefficients of $\mathbf{G}_{\rho u}(z)$ correspond to the $z$ − derivative of the Green's function[19]:

$$G(\boldsymbol{\rho},\mathbf{u},z) = \frac{z e^{ik_c\sqrt{|\mathbf{u} - \boldsymbol{\rho}|^2 + z^2}}}{4\pi(|\mathbf{u} - \boldsymbol{\rho}|^2 + z^2)} \qquad (15)$$

where $k_c$ is the wavenumber at the central frequency. In the Fourier basis ($\mathbf{c} = \mathbf{k}$), $\mathbf{G}_{\rho k}$ simply corresponds to the Fourier transform operator[17]:

$$G(\boldsymbol{\rho},\mathbf{k}) = \exp(j\mathbf{k}.\boldsymbol{\rho}) = \exp\left( j(k_x x + k_y y) \right). \qquad (16)$$

At each depth $z$, the reflected wave-fronts contained in $\mathbf{R}_{\rho c}$ are then decomposed into the sum of a geometric component $\mathbf{G}_{\rho c}$, that

would be ideally obtained in absence of aberrations, and a distorted component that corresponds to the gap between the measured wave-fronts and their ideal counterparts [Fig. 1j][17,19]:

$$\mathbf{D}_{\rho c}(z) = \mathbf{G}_{\rho c}^*(z) \circ \mathbf{R}_{\rho c}(z) \qquad (17)$$

where the symbol $\circ$ stands for a Hadamard product. $\mathbf{D}_{rc} = \mathbf{D}_{\rho c}(z) = [D(\{\boldsymbol{\rho}_{in}, z\}, \mathbf{c}_{out})]$ is the so-called distortion matrix, here expressed at the output. Note that the same operations can be performed by exchanging input and output to obtain the input distortion matrix $\mathbf{D}_{cr} = [D(\mathbf{c}_{in}, \mathbf{r}_{out})] = [D(\mathbf{c}_{in}, \{\boldsymbol{\rho}_{out}, z\})]$.

### Local correlation analysis of the D-matrix

The next step is to exploit local correlations in $\mathbf{D}_{rc}$ to extract the $\mathbf{T}$-matrix. To that aim, a set of output correlation matrices $\mathbf{C}_{out}(\mathbf{r}_p)$ shall be considered between distorted wave-fronts in the vicinity of each point $\mathbf{r}_p$ in the field of view:

$$C(\mathbf{c}_{out}, \mathbf{c}'_{out}, \mathbf{r}_p) = \left\langle D(\mathbf{r}_{in}, \mathbf{c}_{out}) D^*(\mathbf{r}_{in}, \mathbf{c}'_{out}) \mathcal{W}(\mathbf{r}_{in} - \mathbf{r}_p) \right\rangle_{\mathbf{r}_{in}} \qquad (18)$$

An equivalent operation can be performed in input in order to extract a local correlation matrix $\mathbf{C}_{in}(\mathbf{r}_p)$ from the input distortion matrix $\mathbf{D}_{cr}$.

### Iterative phase reversal algorithm

The iterative phase reversal algorithm is a computational process that provides an estimator of the transmission matrix,

$$\mathbf{T}_{out}(z) = \mathbf{G}_{\rho c}^\top(z) \times \mathbf{H}_{out}(z), \qquad (19)$$

where the superscript $\top$ stands for matrix transpose. $\mathbf{T}_{out} = [T(\mathbf{c}_{out}, \mathbf{r}_p)]$ links each point $\mathbf{c}_{out}$ in the dual basis and each voxel $\mathbf{r}_p$ of the medium to be imaged [Fig. 1k]. Mathematically, the algorithm is based on the following recursive relation:

$$\hat{\mathbf{T}}_{out}^{(n)}(\mathbf{r}_p) = \exp\left[ i \arg\left\{ \mathbf{C}_{out}(\mathbf{r}_p) \times \hat{\mathbf{T}}_{out}^{(n-1)}(\mathbf{r}_p) \right\} \right] \qquad (20)$$

where $\hat{\mathbf{T}}_{out}^{(n)}$ is the estimator of $\mathbf{T}_{out}$ at the $n^{th}$ iteration of the phase reversal process. $\hat{\mathbf{T}}_{out}^{(0)}$ is an arbitrary wave-front that initiates the iterative phase reversal process (typically a flat phase law) and $\hat{\mathbf{T}}_{out} = \lim_{n\to\infty} \hat{\mathbf{T}}_{out}^{(n)}$ is the result of this iterative phase reversal process.

This iterative phase reversal algorithm, repeated for each point $\mathbf{r}_p$, yields an estimator $\hat{\mathbf{T}}_{out}$ of the $\mathbf{T}$-matrix. Its digital phase conjugation enables a local compensation of aberrations [Fig. 1l]. The focused $\mathbf{R}$-matrix can be updated as follows:

$$\mathbf{R}_{\rho\rho}^{(corr)}(z) = \left[ \mathbf{D}_{\rho c}(z) \circ \hat{\mathbf{T}}_{out}^\dagger(z) \right] \times \mathbf{G}_{\rho c}^\dagger(z) \qquad (21)$$

where the symbol † stands for transpose conjugate. The same process is then applied to the input correlation matrix $\mathbf{C}_{in}$ for the estimation of the input transmission matrix, $\mathbf{T}_{in}(z) = \mathbf{G}_{\rho c}^\top(z) \times \mathbf{H}_{in}(z)$.

### Multi-scale analysis of wave distortions

To ensure the convergence of the IPR algorithm, several iterations of the aberration correction process are performed while reducing the size of the patches $\mathcal{W}$ with an overlap of 50% between them. Three correction steps are performed in the pork tissue experiment, whereas six are performed in the head phantom experiment [as described in Table 3]. At each step, the correction is performed both at input and output and reciprocity between input and output aberration laws is checked. The correction process is stopped if the normalized scalar product $P_{in/out}$ does not reach 0.9.

**Table 4 | Computational insights**

|  |  | 2D imaging | | 3D imaging | |
|---|---|---|---|---|---|
| Number of channels [Input × Output] |  | $32 \times 32 \approx 10^3$ | | $1024 \times 1024 \approx 10^6$ | |
| Field of view ($\Delta x$, $\Delta y$, $\Delta z$) |  | (20, 0, 80) mm | | (20, 20, 80) mm | |
|  |  | Data | Time | Data | Time |
| Reflection matrix acquisition: $\mathbf{R_{uu}}(t)$ |  | 6 Mo | 8 ms | 6 Go | 260 ms |
| Confocal image $\mathcal{I}(\mathbf{r})$ |  | 53 ko | 5.1 ms | 2.2 Mo | 1.3 min |
| Matrix Imaging | Focused R-matrix: $\mathbf{R_{\rho\rho}}(z)$ | 2.2 Mo | 15 ms | 3.6 Go | 2.3 h |
|  | Estimation of $\mathbf{T}$ & correction |  | 0.15 s |  | 4.5 min |

Here, we compare the typical amount of data and computational time at each post-processing step of UMI. The comparison between 2D and 3D imaging is made using a single line of transducers versus all the transducers of our matrix array. In both cases, the pixel/voxel resolution is fixed at 0.5 mm, which corresponds approximately to one wavelength. The maximum distance between the input and output focusing points is set to 10 mm. The estimation of $\mathbf{T}$ is here investigated without a multi-scale analysis on a single iteration at input and output.

### Synthesize a 1D linear array

To estimate the benefits of 3D imaging compared to 2D UMI, a simulation of a 1D array is performed on experimental ultrasound data acquired with our 2D matrix array. To that aim, cylindrical time delays are applied at input and output:

$$\tau'(\theta^{(s)}, s, z) = \frac{s \sin \theta^{(s)} + z \cos \theta^{(s)}}{c_0} \tag{22}$$

$$\tau'(u^{(s)}, s, z) = \frac{\sqrt{(s - u^{(s)})^2 + z^2}}{c_0}. \tag{23}$$

with $s = x$ or $y$, depending on our focus plane choice.

The focused **R**-matrix is still built in the time domain but using this time the following delay-and-sum beamforming:

$$R^{(2D)}(y_{\text{in}}, y_{\text{out}}, z) = \sum_{\boldsymbol{\theta}_{\text{in}}} \sum_{\mathbf{u}_{\text{out}}} R \left( \boldsymbol{\theta}_{\text{in}}, \mathbf{u}_{\text{out}}, \overbrace{\tau'(\theta_{\text{in}}^{(y)}, y_{\text{in}}, z) + \tau'(u_{\text{out}}^{(y)}, y_{\text{out}}, z)}^{\text{2D beamforming along }(y, z)-\text{plane}} \right.$$
$$\left. + \underbrace{\tau'(\theta_{\text{in}}^{(x)}, x_{\text{f}}, z_{\text{f}}) + \tau'(u_{\text{out}}^{(x)}, x_{\text{f}}, z_{\text{f}}) - 2z_{\text{f}}/c_0}_{\text{Cylindrical law to focus at } (x_{\text{f}}, z_{\text{f}})} \right). \tag{24}$$

The images displayed in Fig. 6b, c are obtained by synthesizing input and output beams collimated in the $(y, z)$ – plane by focusing on a line located at ($x_{\text{f}} = 0$ mm, $z_{\text{f}} = 37.25$ mm), thereby mimicking the beamforming process by a conventional linear array of transducers.

### Estimation of contrast and resolution

Contrast and resolution are evaluated by means of the RPSF. Equivalent to the full width at half maximum commonly used in 2D UMI, the transverse resolution $\delta\rho$ is assessed in 3D based on the area $\mathcal{A}_{(-3\,\text{dB})}$ at half maximum of the RPSF amplitude:

$$\delta\rho_{(-3dB)} = \sqrt{\mathcal{A}_{(-3dB)}/\pi} \tag{25}$$

The contrast, $\mathcal{F}$, is computed locally by decomposing the normalized RPSF as the sum of three components[28]:

$$\overline{RPSF}(\mathbf{r}_{\text{p}}, \Delta\boldsymbol{\rho}) = \frac{RPSF(\mathbf{r}_{\text{p}}, \Delta\boldsymbol{\rho})}{RPSF(\mathbf{r}_{\text{p}}, \Delta\boldsymbol{\rho} = \mathbf{0})} = \alpha_S(\mathbf{r}_{\text{p}}) + \alpha_M(\mathbf{r}_{\text{p}}) + \alpha_N(\mathbf{r}_{\text{p}}). \tag{26}$$

$\alpha_S$ is the single scattering rate that corresponds to the confocal peak. $\alpha_M$ is a multiple scattering rate that gives rise to a diffuse halo; $\alpha_N$ corresponds to the electronic noise rate which results in a flat plateau. A local contrast can then be deduced from the ratio between $\alpha_S$ and the incoherent background $\alpha_B = \alpha_M + \alpha_N$,

$$\mathcal{F}(\mathbf{r}_{\text{p}}) = \frac{\alpha_S(\mathbf{r}_{\text{p}})}{\alpha_B(\mathbf{r}_{\text{p}})} = \frac{1 - \alpha_B(\mathbf{r}_{\text{p}})}{\alpha_B(\mathbf{r}_{\text{p}})} \tag{27}$$

### Single and multiple scattering rates

The single scattering, multiple scattering and noise rates can be directly computed from the decomposition of the RPSF (Eq. (26)). However, at large depths, multiple scattering and noise are difficult to discriminate since they both give rise to a flat plateau in the RPSF. In that case, the spatial reciprocity symmetry can be invoked to differentiate their contribution. The multiple scattering component actually gives rise to a symmetric **R**-matrix while electronic noise is associated with a fully random matrix. The relative part of the two components can thus be estimated by computing the degree of anti-symmetry $\beta$ in the **R**-matrix. To that aim, the **R**-matrix is first projected onto its anti-symmetric subspace at each depth:

$$\mathbf{R}_{\boldsymbol{\rho\rho}}^{(A)}(z) = \frac{\mathbf{R}_{\boldsymbol{\rho\rho}}(z) - \mathbf{R}_{\boldsymbol{\rho\rho}}^{\top}(z)}{2} \tag{28}$$

where the superscript $\top$ stands for matrix transpose. In a common midpoint representation, (Eq. (28)) re-writes:

$$R_{\mathcal{M}}^{(A)}(\mathbf{r}_{\text{m}}, \Delta\boldsymbol{\rho}) = \frac{R_{\mathcal{M}}(\mathbf{r}_{\text{m}}, \Delta\boldsymbol{\rho}) - R_{\mathcal{M}}(\mathbf{r}_{\text{m}}, -\Delta\boldsymbol{\rho})}{2}. \tag{29}$$

A local degree of anti-symmetry $\beta$ is then computed as follows:

$$\beta(\mathbf{r}_{\text{p}}) = \frac{\left\langle \left| R_{\mathcal{M}}^{(A)}(\mathbf{r}_{\text{m}}, \Delta\boldsymbol{\rho}) \right|^2 \mathcal{W}(\mathbf{r}_{\text{m}} - \mathbf{r}_{\text{p}}) \mathcal{D}(\Delta\boldsymbol{\rho}) \right\rangle_{[\mathbf{r}_{\text{m}}, \Delta\boldsymbol{\rho}]}}{\left\langle \left| R_{\mathcal{M}}(\mathbf{r}_{\text{m}}, \Delta\boldsymbol{\rho}) \right|^2 \mathcal{W}(\mathbf{r}_{\text{m}} - \mathbf{r}_{\text{p}}) \mathcal{D}(\Delta\boldsymbol{\rho}) \right\rangle_{[\mathbf{r}_{\text{m}}, \Delta\boldsymbol{\rho}]}} \tag{30}$$

where $\mathcal{D}(\Delta\boldsymbol{\rho})$ is a de-scanned window function that eliminates the confocal peak such that the computation of $\beta$ is only made by considering the incoherent background. Typically, we chose $\mathcal{D}(\Delta\boldsymbol{\rho}) = 1$ for $\Delta\boldsymbol{\rho} > 6\delta\rho_0(z)$, and zero otherwise. Assuming equi-partition of the electronic noise between its symmetric and anti-symmetric subspace, the multiple scattering rate $\alpha_M$ and noise ratio $\alpha_N$ can then be deduced (see Supplementary Section 11):

$$\alpha_M(\mathbf{r}_{\text{p}}) = \left( 1 - 2\beta(\mathbf{r}_{\text{p}}) \right) \alpha_B(\mathbf{r}_{\text{p}}) \tag{31}$$

$$\alpha_N(\mathbf{r}_{\text{p}}) = 2\beta(\mathbf{r}_{\text{p}}) \alpha_B(\mathbf{r}_{\text{p}}) \tag{32}$$

In the head phantom experiment [Fig. 5b], these rates are estimated at each depth by averaging over a window of size $\mathbf{w} = (w_\rho, w_z) = (20, 5.5)$ mm.

### Computational insights

While the UMI process is close to real-time for 2D imaging (*i.e.* for linear, curve or phased array probes), 3D UMI (using a fully populated matrix array of transducers) is still far from it (see Table 4) as it involves the processing of much more ultrasound data. Even if computing a confocal 3D image only requires a few minutes, building the focused **R**-matrix from the raw data takes a few hours

(on GPU with CUDA language) while one step of aberration correction only lasts for a few minutes. All the post-processing was realized with Matlab (R2021a) on a working station with 2 processors @2.20GHz, 128Go of RAM, and a GPU with 48 Go of dedicated memory.

## Reporting summary

Further information on research design is available in the Nature Portfolio Reporting Summary linked to this article.

## Data availability

The ultrasound data generated in this study is available at Zenodo[60] (https://zenodo.org/record/8159177).

## Code availability

Codes used to post-process the ultrasound data within this paper are available from the corresponding author upon request.

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

## Acknowledgements

The authors wish to thank L. Marsac for providing initial ultrasound acquisition sequences. The authors are grateful for the funding provided by the European Research Council (ERC) under the European Union's Horizon 2020 research and innovation program (grant agreement 819261, REMINISCENCE project, AA).

## Author contributions

A.A. and M.F. initiated the project. A.A. supervised the project. F.B. and A.L.B. coded the ultrasound acquisition sequences. F.B. and J.R. performed the experiments. F.B., A.L.B., and W.L. developed the post-processing tools. F.B., J.R., and A.A. analyzed the experimental results. A.A. performed the theoretical study. F.B. prepared the figures. F.B., J.R., and A.A. prepared the manuscript. F.B., J.R., A.L.B., W.L., M.F., and A.A. discussed the results and contributed to finalizing the manuscript.

## Competing interests

A.A., M.F., and W.L. are inventors of a patent related to this work held by CNRS (no. US11346819B2, published May 2022). W.L. had his PhD funded by the SuperSonic Imagine company and is now an employee of this company. All authors declare that they have no other competing interests.
