## [Peer Review File · Nature Communications]

Three-Dimensional Ultrasound Matrix ImagingREVIEWER COMMENTS

Reviewer #1 (Remarks to the Author):

The manuscript “Three-Dimensional Ultrasound Matrix Imaging” develops a matrix imaging approach for three-dimensional ultrasound imaging based on an extension of previous two-dimensional results.

Matrix ultrasound imaging is a powerful tool that describes the information captured by an ultrasound system based on a response matrix approach and an analysis of its characteristics, particularly scattering and aberration, to quantify and correct for image-degrading effects. For ultrasound imaging, these are vital questions that have been persistently challenging. Correcting for aberration and reducing reverberation effects are the key components required to extend the capabilities of ultrasound imaging, especially to areas that are currently difficult or impossible to image, such as deep abdominal imaging or transcranial imaging. This topic is therefore highly significant to ultrasound imaging and resolving these challenges would dramatically improve image quality and the overall capabilities of clinical and pre-clinical ultrasound systems as well as emerging methods such as functional ultrasound or super-resolution ultrasound.

In the manuscript, analogies are appropriately drawn to the fields of seismology and astronomy. Indeed, as the authors point out there are significant similarities and differences. However, on critical difference, motion in either in the tissue or by the operator holding the probe occurs far too late in the manuscript. The limitations imposed by motion, especially given the potentially onerous acquisition times, merits treatment in the introduction.

It may initially seem like the extended an existing 2D method to 3D represents a minor achievement. However, the authors convincingly demonstrate that the extension to 3D generates dramatic improvements in image quality, including contrast and reverberation. Therefore, even though some of the theoretical exposition is a somewhat straightforward extension of previous 2D publications, the exciting experimental results, leave no doubt that a 3D implementation has entirely new and more dramatic consequences on improvements in contrast for both the pork muscle and transcranial demonstrations. Furthermore, compared to previous publications, the manuscript contains substantially different and new implementation details and analyses that are performed on the 3D data. Consequently, the experimental results are highly innovative and the analytical treatment also contains significant elements of innovation.

It should be noted that the quality of the experimental demonstration and its cohesiveness in illustrating the proposed matrix imaging approach is excellent. Figs. 2 and 4 in particular, capture and illustrate the strengths of the technique in reducing aberration artifacts, e.g. point spread function doubling, and reverberation artifacts, e.g. contrast enhancements. Fundamental claims about aberration reduction and reverberation suppression are further buttressed and quantified by Fig. 5. These demonstrations therefore clearly support the claims of image quality improvement made in the manuscript.

The methods are described densely but clearly and appear to be sufficient for a skilled user to

reproduce. However, given the complexity of some of the processing techniques, it is difficult to assess whether a full reproduction of the work is feasible without a significant coding implementation effort. Raw ultrasound data is shared, which is commendable, although code is only available upon request. I would encourage the authors to share the implementation code, if possible.

There are a few minor concerns, detailed below. Nevertheless, enthusiasm for this manuscript remains high. It combines a significant extension of previous 2D work to 3D. Challenging experimental demonstrations effectively demonstrate the capabilities of ultrasound matrix imaging. The scientific approach is clearly described. Overall the manuscript is excellent.

A brief discussion of reverberation in the pork muscle of Figs 2 and 4 may help readers understand why artifacts are removed in one case but not the other.

Replace colloquial “pork chop” with pork muscle identifying the anatomical location from which it was obtained.

The “pork chop” is somewhat thin compared to human abdominal wall thickness which is significantly degrading in the 2-3 cm range. It would benefit the readers to compare the aberration of the pork model to standard human measurements, such as “Measurements of ultrasonic pulse distortion produced by human chest wall” Laura M. Hinkelman; Thomas L. Szabo; Robert C. Waag

The discussion of the convergence of T implies that T is converging to the correct solution, which is not the case, it is converging to a consistent representation between transmit and receive representation. Furthermore validating the aberration estimates could be performed experimentally in an independent fashion (e.g. water tank measurements). Further clarifying convergence would be helpful.

Reviewer #2 (Remarks to the Author):

This paper describes the adaptation of the ultrasound matrix imaging (UMI) approach to 3D ultrasound imaging. The paper largely adapts the methods described in references [18] and [27] to volumetric imaging, although provides additional insightful analyses to support aspects of the approach that were not featured in [18] and [27]. I did not have any major concerns over the science in this manuscript, as much of it was well supported, and enjoyed reading the paper and delving into the concepts of the paper. The paper is a more challenging read than the average paper in this field due to the mathematical and theoretical concepts used. I admit that I had to rely substantially on references [18] and [27] to follow some of the concepts in this paper. For example, the description of the use of common midpoints (CMP) in reference [18] was helpful and informative to understanding the volumetric RPSF concept in this paper. Perhaps there wasn't an easy way to describe the same idea as a CMP for the virtual transmitter and receiver sources in an xy-plane.

The paper follows the technical format of the method in [18] and [27], but the authors do not explain where any of the adapted methodology might diverge, if at all, from the prior work (aside from the 3D nature of the adaptation). I would agree, however, that the experiments and data analysis are more difficult than standard aberration correction on linear arrays, and the results are indeed impressive. The authors would probably need to better clarify any differences in this work over the authors prior publications on involving linear arrays (aside from the 3D adaptation). The main conclusion of the paper appears to be that the 3D version yields a more accurate estimate and correction of the aberration law, which is true, but this idea has also been well established since the early 90's, so this conclusion was expected. The authors appear to overlook the prior literature on the need for 2D estimation and correction of aberration in their introduction (see, for example, papers from the laboratories of Matt O'Donnell, Gregg Trahey, Mathias Fink, Steve Smith, and Robert Waag).

More specific comments on the paper are:

1. Lines 68-71: I think that *technically* this sentence is correct, but it really does leave the reader the impression that you will demonstrate improvement in ultrafast Doppler and ULM later in the paper. My suggestion is to avoid mentioning these techniques here (you do discuss these later, which is okay).

2. Lines 184-203: The authors describe an evaluation of the "convergence" of the aberration estimate, \hat{T}_{in} , but they do not explicitly describe what they mean by convergence right away and it appears that there are two definitions of convergence being used. My understanding is that the purpose of this section is to show that the aberration estimation is accurate when \hat{T}_{in} is roughly equivalent to \hat{T}_{out} . My major concern is that I don't agree that $T_{in}^{\hat{T}_{in}}$ and $\hat{T}_{in}^{\hat{T}_{out}}$ can be compared or equated, as these two functions represent different convergences. The $T_{in}^{\hat{T}_{in}}$ function represents how well the estimate \hat{T}_{in} , as a function of N_w , matches the reference aberration law created by computing T_{in} with $N_w=100$. $T_{in}^{\hat{T}_{in}}$ converges because, as N_w increases, the representative data from which \hat{T}_{in} is computed becomes increasingly identical to the data from which T_{in} was computed. $\hat{T}_{in}^{\hat{T}_{out}}$ compares how well \hat{T}_{in} is matched to \hat{T}_{out} as a function of N_w . Theoretically, $\hat{T}_{in}^{\hat{T}_{out}}$ should be close to 1 for all of N_w , however, this function starts low and converges as N_w increases due to the various noises (multiple-scattering, random noise). In fact, the authors show this in Fig. S6, where the reciprocity between \hat{T}_{in} and \hat{T}_{out} can be low due to noise. Therefore, even though Fig. 3d shows a close match between the curves, I don't think these curves can be related.

3. In Fig. 3 (and sometimes seen elsewhere, such as S2 & S3), there is significant phase wrapping shown in the aberration phase laws. This wrapping is indicated by the transition from red-to-black-to-blue or blue-to-black-to-red. Phase wrapping normally creates issues with aberration correction. Can the authors either explain why this method is not affected by this phase wrapping or how they unwrap the phase for correction?

4. Line 240-241: It is unclear what the authors were saying here (there could be a typo here, and I'm not sure what was trying to be said).

5. Line 312-313: What do you mean by more efficient? Supplementary Section S3 did not describe efficiency.

6. Line 315: "adaptive" seems out of place here. Did you mean to say local or individual aberration phase law?

7. Line 327-328: I'm glad you included this and section S8 (and Fig. S7), because I think that there are consequences for "undersampled" transmissions, like that used in ultrafast imaging. Because the ultrafast transmissions are rather popular in research, it is important for readers to understand the implications here. I would probably also add that the "undersampled" plane wave transmissions will likely create a problem in having T_{in} and T_{out} converge since the underlying "PSF"s diverge significantly. In this paper, because the plane wave transmissions were highly sampled (1225 plane wave transmissions), there should be good agreement between the in and out "PSF"s.

8. Line 433: "In the transducer basis ($\omega = k$),..." Should that say " $\omega = u$ " instead?

9. Line 443: Is the assumption correct that D_{ri} means $D(\rho_{out}, \omega_{in}, z)$? If so, I find the notation here a bit hard to follow/read.

10. In the discussion of N_w as shown in Eq. 6, it was stated that the bias scales inversely proportional to N_w . I was never able to fully follow the discussion in the paper and the supplementary material on that, but I did find (through the authors prior publications and not this one!) that reference [15] help greatly. In the discussions in the paper and supplementary material, this reference was not cited, but rather other citations that did not seem to address this topic.

In the following we will provide a detailed response (in blue) to the reports of Referees (shown in italic black):

A. Introduction

We would like to thank the referees for their thoughtful feedback. Below we submit our answers to the referee comments and questions, and detail the changes made to the manuscript in response to the referee reports.

B. Response to Referee Reports

In the following, we answer the Referees' concerns and describe changes made to the manuscript in response to their comments. Referee comments are given in italics, followed by our responses in blue. Note that numbered references refer to the current (newly resubmitted) version of the manuscript.

Referee 1

The manuscript “Three-Dimensional Ultrasound Matrix Imaging” develops a matrix imaging approach for three-dimensional ultrasound imaging based on an extension of previous two-dimensional results.

Matrix ultrasound imaging is a powerful tool that describes the information captured by an ultrasound system based on a response matrix approach and an analysis of its characteristics, particularly scattering and aberration, to quantify and correct for image-degrading effects. For ultrasound imaging, these are vital questions that have been persistently challenging. Correcting for aberration and reducing reverberation effects are the key components required to extend the capabilities of ultrasound imaging, especially to areas that are currently difficult or impossible to image, such as deep abdominal imaging or transcranial imaging. This topic is therefore highly significant to ultrasound imaging and resolving these challenges would dramatically improve image quality and the overall capabilities of clinical and pre-clinical ultrasound systems as well as emerging methods such as functional ultrasound or super-resolution ultrasound.

We would like to express our gratitude for the thorough revision and constructive comments provided by the reviewer. Their feedback was extremely relevant and helped us to improve the quality and clarity of our research.

In the manuscript, analogies are appropriately drawn to the fields of seismology and astronomy. Indeed, as the authors point out there are significant similarities and differences. However, on critical difference, motion in either in the tissue or by the operator holding the probe occurs far too late in the manuscript. The limitations imposed by motion, especially given the potentially onerous acquisition times, merits treatment in the introduction.

We agree that we did not insist enough on this particular feature of US imaging. The problem of motion during acquisition is now clearly stated in in the introduction of the manuscript:

- Page 2, lines 32-35: “Such a complex adaptive focusing scheme cannot be implemented in real time since it is extremely sensitive to motion [1] whether induced by the operator holding the

probe or by the movement of tissues.”

We now also stress on the major assumption of a medium considered as static during the acquisition of reflection matrix:

- Page 3, lines 42-43: “ More specifically, assuming that the medium remains fixed during the acquisition...”

It may initially seem like the extended an existing 2D method to 3D represents a minor achievement. However, the authors convincingly demonstrate that the extension to 3D generates dramatic improvements in image quality, including contrast and reverberation. Therefore, even though some of the theoretical exposition is a somewhat straightforward extension of previous 2D publications, the exciting experimental results, leave no doubt that a 3D implementation has entirely new and more dramatic consequences on improvements in contrast for both the pork muscle and transcranial demonstrations. Furthermore, compared to previous publications, the manuscript contains substantially different and new implementation details and analyses that are performed on the 3D data. Consequently, the experimental results are highly innovative and the analytical treatment also contains significant elements of innovation.

It should be noted that the quality of the experimental demonstration and its cohesiveness in illustrating the proposed matrix imaging approach is excellent. Figs. 2 and 4 in particular, capture and illustrate the strengths of the technique in reducing aberration artifacts, e.g. point spread function doubling, and reverberation artifacts, e.g. contrast enhancements. Fundamental claims about aberration reduction and reverberation suppression are further buttressed and quantified by Fig. 5. These demonstrations therefore clearly support the claims of image quality improvement made in the manuscript.

The methods are described densely but clearly and appear to be sufficient for a skilled user to reproduce. However, given the complexity of some of the processing techniques, it is difficult to assess whether a full reproduction of the work is feasible without a significant coding implementation effort. Raw ultrasound data is shared, which is commendable, although code is only available upon request. I would encourage the authors to share the implementation code, if possible.

In this resubmission, we have included our post-processing codes such that the reviewers can have access to them. With regards to open source, we are aware that sharing the implementation code would be beneficial to the community. However, this work has been done in collaboration with the company Hologic / SuperSonic Imagine which generates issues about intellectual property and open science. We are currently in discussion with this company in order to see which part of the code we can share as open source. At this stage, we did not get a green light yet, but this will be done hopefully before publication of this work.

There are a few minor concerns, detailed below. Nevertheless, enthusiasm for this manuscript remains high. It combines a significant extension of previous 2D work to 3D. Challenging experimental demonstrations effectively demonstrate the capabilities of ultrasound matrix imaging. The scientific approach is clearly described. Overall the manuscript is excellent.

We are extremely pleased with the referee’s enthusiasm for this work.

1/ *A brief discussion of reverberation in the pork muscle of Figs 2 and 4 may help readers understand why artifacts are removed in one case but not the other.*

There are no reverberations in the pork tissue experiment since the impedance mismatch with the ultrasonic phantom is much less important than in the skull experiment. This is now clearly stated in the “Discussion” section:

- Page 20, lines 375-380: “In addition to aberrations, another issue for transcranial imaging arises from multiple reflections caused by the skull. While such reverberations are not observed in the pork tissue experiment, their detrimental effects are much greater in a transcranial experiment because of the large impedance mismatch between the skull and brain tissues. In this work, such artefacts are not corrected and they drastically pollute the image at shallow depths ($z < 20$ mm).”

2/ *Replace colloquial “pork chop” with pork muscle identifying the anatomical location from which it was obtained.*

We agree with the reviewer. In the new version of the manuscript, we have removed all the “pork chop” but now use the term “pork tissue” rather than “pork muscle”. Indeed it is composed of both fat and muscle tissue, and this mixture is actually the cause of aberrations in our experiment.

3/ *The “pork chop” is somewhat thin compared to human abdominal wall thickness which is significantly degrading in the 2-3 cm range. It would benefit the readers to compare the aberration of the pork model to standard human measurements, such as “Measurements of ultrasonic pulse distortion produced by human chest wall” Laura M. Hinkelman; Thomas L. Szabo; Robert C. Waag.*

In the most aberrated area (under the fat), the root mean square (rms) of the extracted time delay law is of 56 ns. In the new version of the manuscript, a comparison of this value with previous measurements of Hinkelman et al. through an abdominal wall section (average rms of 43 ns). Both values are in the same order of magnitude, which indicates that the pork tissues used in our experiment give rise to a level of aberrations comparable with standard human measurements:

- Page 13, lines 232-237: “In the most aberrated area (*i.e* under the fat), the phase fluctuations of the aberration law corresponds to a time delay spread of 56 ns (rms). This value is comparable with past measurements through the human abdominal wall [2]. The pork tissue layer thus induces a level of aberrations typical of standard ultrasound diagnosis.”

4/ *The discussion of the convergence of \mathbf{T} implies that \mathbf{T} is converging to the correct solution, which is not the case, it is converging to a consistent representation between transmit and receive representation. Furthermore validating the aberration estimates could be performed experimentally in an independent fashion (e.g. water tank measurements). Further clarifying convergence would be helpful.*

The “Spatial reciprocity as a guide star” section of the accompanying paper was revised to answer this specific question, which was also raised by reviewer #2.

First, the convergence of $\hat{\mathbf{T}}_{\text{in}}$ is no longer investigated by considering its value for large window size as the ground truth. We agree with the reviewer that this method was biased. The corresponding curves have been removed from Figs. 3a and b. Instead, The validity of the \mathbf{T} -matrix estimator is now checked by examining the effect of the correction on the RPSFs in a neighbouring region \mathcal{W}_2 that lies directly below the area \mathcal{W}_1 where we extract the aberration law (see new Fig.3c). Applying

the aberration correction on a region different from where it was estimated allows us to avoid any overfitting bias. The following changes have been made in the manuscript:

- Page 12, Figure 3: New panels g and h have been included to show the evolution of the RPSF in \mathcal{W}_2 when phase laws displayed in panels e and f are used to compensate for aberrations. The caption of Fig.3 has been modified accordingly.
- Page 10-11, lines 190-197: “The validity of the \mathbf{T} –matrix estimator in a region \mathcal{W}_1 (Fig. 3c) is investigated by examining the corrected RPSF in a neighbour region \mathcal{W}_2 (yellow box). \mathcal{W}_1 and \mathcal{W}_2 are sufficiently close to assume, in a first approximation, that they belong to the same isoplanatic patch. If the box is too small (Fig. 3d₁), our estimator has not converged yet and the correction is not valid, as shown by the degraded quality of the RPSF in \mathcal{W}_2 [Fig. 3h₁] compared with its initial value [Fig. 3g]. With sufficient spatial averaging [Fig. 3d₃], a valid aberration law can be extracted as shown by a corrected RPSF close to be only diffraction-limited [Fig. 3h₃].”

Following these changes, we have also modified the paragraph showing the scalar product $P_{\text{in/out}}$ (new notation) between $\hat{\mathbf{T}}_{\text{in}}$ and $\hat{\mathbf{T}}_{\text{out}}$ as a relevant observable for monitoring the convergence of the \mathbf{T} –matrix estimator:

- Page 11, lines 198-211: “The question that now arises is how we can, in practice, know if the convergence of $\hat{\mathbf{T}}$ is fulfilled without any *a priori* knowledge on \mathbf{T} . An answer can be found by comparing the estimated input and output aberration phase laws, $\hat{T}_{\text{in}}(\mathbf{u}, \mathbf{r}_p)$ and $\hat{T}_{\text{out}}(\mathbf{u}, \mathbf{r}_p)$, at a given point \mathbf{r}_p as shown in Figs. 3e and f. Spatial reciprocity implies that \hat{T}_{in} and \hat{T}_{out} shall be equal when the convergence of the estimator is reached [Figs. 3e₃,f₃]. Their normalized scalar product, $P_{\text{in/out}} = N_u^{-1} \hat{\mathbf{T}}_{\text{in}} \hat{\mathbf{T}}_{\text{out}}^\dagger$, can be used to probe the error made on the aberration phase law $|\delta T|^2$. Both quantities are actually related as follows (see Supplementary Section S7):

$$|\delta T|^2 \simeq 1 - P_{\text{in/out}}. \quad (1)$$

The normalized scalar product $P_{\text{in/out}}$ is displayed as a function of w and shows the convergence of the IPR process [Fig. 3a]. For a sufficiently large box [Fig. 3d₃], $\hat{\mathbf{T}}$ is supposed to have converged towards \mathbf{T} when $\hat{\mathbf{T}}_{\text{in}}$ and $\hat{\mathbf{T}}_{\text{out}}$ are almost equal [Fig. 3e₃,f₃], while, for a small box [Fig. 3d₁], a large discrepancy can be found between them. In the following, the parameter $P_{\text{in/out}}$ will thus be used as a guide star for monitoring the convergence of the UMI process.”

The impact of the imaging dimension on the convergence has also been developed to clarify our demonstration:

- Page 12, Figure 3a: The normalized scalar product $P_{\text{in/out}}$ (new notation) is now displayed as a function of the window size w , expressed in mm, instead of $N_{\mathcal{W}}$, expressed in resolution cells, to better illustrate the superiority of 3D over 2D imaging.
- Pages 11-12, lines 212-219: “The scaling law of Eq. 6 with respect to $N_{\mathcal{W}}$ is checked in Fig. 3b. The inverse scaling of the bias with $N_{\mathcal{W}}$ shows the advantage of 3D UMI over 2D UMI, since $N_{\mathcal{W}} \sim w^d$, with d the imaging dimension. This superiority is evident in Fig. 3a, which shows a faster convergence with 3D boxes (green curve) than with 2D patches (orange curve). For a given precision, 3D UMI thus provides a better spatial resolution for our \mathbf{T} –matrix estimator as shown by Figs. 3f₃ and f₄, where much better agreement between $\hat{\mathbf{T}}_{\text{in}}$ and $\hat{\mathbf{T}}_{\text{out}}$ is observed for a 3D box [Fig. 3d₃] than for a 2D patch [Fig. 3d₄] of same dimension w .”

Referee 2

This paper describes the adaptation of the ultrasound matrix imaging (UMI) approach to 3D ultrasound imaging. The paper largely adapts the methods described in references [18] and [27] to volumetric imaging, although provides additional insightful analyses to support aspects of the approach that were not featured in [18] and [27]. I did not have any major concerns over the science in this manuscript, as much of it was well supported, and enjoyed reading the paper and delving into the concepts of the paper.

We are glad that the reviewer enjoyed reading the paper. We thank them for the constructive comments that helped us to improve the level of the manuscript. We hope the referee will be convinced by the changes we made to the manuscript.

1/ The paper is a more challenging read than the average paper in this field due to the mathematical and theoretical concepts used. I admit that I had to rely substantially on references [18] and [27] to follow some of the concepts in this paper. For example, the description of the use of common midpoints (CMP) in reference [18] was helpful and informative to understanding the volumetric RPSF concept in this paper. Perhaps there wasn't an easy way to describe the same idea as a CMP for the virtual transmitter and receiver sources in an xy -plane.

Yes the reviewer is right, the common mid-point basis was not properly introduced in the manuscript. In response to the reviewer, the introduction to the RPSF is now done in two steps to be more didactic: (i) change of variables to a common midpoint representation where the reflection matrix is referred to as $\mathbf{R}_{\mathcal{M}}$; (ii) the ‘‘RPSF’’ is now defined as the average intensity of $\mathbf{R}_{\mathcal{M}}$:

- Page 8, lines 122-133: ‘‘Such an operation is mathematically equivalent to a change of variable to express the focused \mathbf{R} -matrix in a common midpoint basis [1] (see also Supplementary Section S2):

$$R_{\mathcal{M}}(\mathbf{r}_m, \Delta\rho) = R\left(\rho_m - \frac{\Delta\rho}{2}, \rho_m + \frac{\Delta\rho}{2}, z\right),$$

where the subscript \mathcal{M} stands for the common midpoint basis. $\mathbf{r}_m = \{\rho_m, z\} = \{(\rho_{\text{in}} + \rho_{\text{out}})/2, z\}$ is the common midpoint between the input and output focal spots, with the two separated by a distance $\Delta\rho = \rho_{\text{out}} - \rho_{\text{in}}$. In the speckle regime (random reflectivity), this quantity probes the local focusing quality as its ensemble average intensity, which we refer to as the ‘‘reflection point spread function (RPSF)’’, scales as an incoherent convolution between the input and output PSFs [1]:

$$RPSF(\mathbf{r}, \Delta\rho) = \langle |R_{\mathcal{M}}(\mathbf{r}, \Delta\rho)|^2 \rangle \propto |H_{\text{in}}|^2 \otimes |H_{\text{out}}|^2(\mathbf{r}, \Delta\rho), \quad (2)$$

where $\langle \dots \rangle$ denotes an ensemble average, which, in practice, is performed by a local spatial average (see Methods).’’

- A new Supplementary Section S2 and new Figure S2 have also been included to illustrate the different steps required to extract the RPSF, first using a 2D imaging configuration before generalising this approach to 3D imaging.

2/ The paper follows the technical format of the method in [18] and [27], but the authors do not explain where any of the adapted methodology might diverge, if at all, from the prior work (aside from

the 3D nature of the adaptation). I would agree, however, that the experiments and data analysis are more difficult than standard aberration correction on Linear arrays, and the results are indeed impressive. The authors would probably need to better clarify any differences in this work over the authors prior publications on involving Linear arrays (aside from the 3D adaptation).

To better explain the differences from our previous work, we have rewritten the beginning of the “Discussion” section in order to outline the novel aspects of this work compared with previous studies. Apart from the 3D imaging feature, the main differences in this work are the introduction of the IPR algorithm, which outperforms the SVD, and the introduction of the spatial reciprocity criterion to check the convergence of our \mathbf{T} estimate. The following paragraph has been included in the new version of the manuscript:

- Page 18-19, lines 323-338: “This work is not only a 3D extension of previous studies [17,28]. Indeed, several crucial elements have been introduced to make UMI more robust. First, the proposed iterative phase reversal algorithm outperforms the SVD for local compensation of aberrations because it can evaluate the aberration law on a larger angular support (see Supplementary Section S5), resulting in a faster correction process. Second, the bias of our \mathbf{T} -matrix estimator has been expressed analytically (Eq. 6) as a function of the coherence factor that grasps the detrimental effects of the virtual guide star blurring induced by aberrations, multiple scattering and noise. This led us to define a general strategy for UMI with: (i) a multi-scale compensation of wave distortions to gradually reduce the blurring of the virtual guide star and tackle high-order aberrations associated with small isoplanatic lengths; (ii) the application of an adaptive confocal filter to cope with multiple scattering and noise; (iii) a fine monitoring of the convergence of our estimator by mean of spatial reciprocity. The latter is a real asset as it provides an objective criterion to check the physical significance of the extracted aberration laws and optimize the resolution of our \mathbf{T} -matrix estimator.”

3/ The main conclusion of the paper appears to be that the 3D version yields a more accurate estimate and correction of the aberration law, which is true, but this idea has also been well established since the early 90’s, so this conclusion was expected. The authors appear to overlook the prior literature on the need for 2D estimation and correction of aberration in their introduction (see, for example, papers from the laboratories of Matt O’Donnell, Gregg Trahey, Mathias Fink, Steve Smith, and Robert Waag).

The reviewer is absolutely right, we should have cited these works. We now mention these studies in the introduction (see references [29-32] below):

- Page 3, line 52-56: “Yet, aberrations in the human body are 3D-distributed and a 1D control of the wave-field is not sufficient for a fine compensation of wave-distortions as already shown by previous works [29-32].”
- [29] N. M. Ivancevich, J. J. Dahl, G. E. Trahey, and S. W. Smith, Phase-aberration correction with a 3-D ultrasound scanner: Feasibility study, IEEE Trans. Ultrason. Ferroelectr. Freq. Control 53, 1432 (2006).
- [30] J. Lacefield and R. Waag, Time-shift estimation and focusing through distributed664 aberration using multirow arrays, IEEE Trans. Ultrason. Ferroelectr. Freq. Control 48, 1606 (2001).

- [31] B. D. Lindsey and S. W. Smith, Pitch-catch phase aberration correction of multiple isoplanatic patches for 3-D transcranial ultrasound imaging, IEEE Trans. Ultrason. Ferroelectr. Freq. Control 60, 463 (2013).
- [32] D.-L. Liu and R. Waag, Estimation and correction of ultrasonic wavefront distortion using pulse-echo data received in a two-dimensional aperture, IEEE Trans. Ultrason. Ferroelectr. Freq. Control 45, 473 (1998).

More specific comments on the paper are:

4/ Lines 68-71: I think that **technically** this sentence is correct, but it really does leave the reader the impression that you will demonstrate improvement in ultrafast Doppler and ULM later in the paper. My suggestion is to avoid mentioning these techniques here (you do discuss these later, which is okay).

Following the reviewer’s remark, we have removed that part of the introduction and kept the comments on that aspect for discussion.

5/ Lines 184-203: The authors describe an evaluation of the “convergence” of the aberration estimate, $\hat{\mathbf{T}}_{in}$, but they do not explicitly describe what they mean by convergence right away and it appears that there are two definitions of convergence being used. My understanding is that the purpose of this section is to show that the aberration estimation is accurate when $\hat{\mathbf{T}}_{in}$ is roughly equivalent to $\hat{\mathbf{T}}_{out}$. My major concern is that I don’t agree that $(\hat{\mathbf{T}}_{in} | \mathbf{T}_{in})$ and $(\hat{\mathbf{T}}_{in} | \hat{\mathbf{T}}_{out})$ can be compared or equated, as these two functions represent different convergences. The $(\hat{\mathbf{T}}_{in} | \mathbf{T}_{in})$ function represents how well the estimate $\hat{\mathbf{T}}_{in}$, as a function of $N_{\mathcal{W}}$, matches the reference aberration law created by computing \mathbf{T}_{in} with $N_{\mathcal{W}} = 100$. $(\hat{\mathbf{T}}_{in} | \mathbf{T}_{in})$ converges because, as $N_{\mathcal{W}}$ increases, the representative data from which $\hat{\mathbf{T}}_{in}$ is computed becomes increasingly identical to the data from which \mathbf{T}_{in} was computed. $(\hat{\mathbf{T}}_{in} | \hat{\mathbf{T}}_{out})$ compares how well $\hat{\mathbf{T}}_{in}$ is matched to $\hat{\mathbf{T}}_{out}$ as a function of $N_{\mathcal{W}}$. Theoretically, $(\hat{\mathbf{T}}_{in} | \hat{\mathbf{T}}_{out})$ should be close to 1 for all of $N_{\mathcal{W}}$, however, this function starts low and converges as $N_{\mathcal{W}}$ increases due to the various noises (multiple-scattering, random noise). In fact, the authors show this in Fig. S6, where the reciprocity between $\hat{\mathbf{T}}_{in}$ and $\hat{\mathbf{T}}_{out}$ can be low due to noise. Therefore, even though Fig. 3d shows a close match between the curves, I don’t think these curves can be related.

The “Spatial reciprocity as a guide star” section of the accompanying paper has been revised to answer this issue, which was also raised by reviewer #1.

First, the convergence of $\hat{\mathbf{T}}_{in}$ is no longer investigated by considering its value for large window size as the ground truth. We agree with the reviewer that this method was biased. The corresponding curves have been removed from Fig.3a and b. Instead, The validity of the \mathbf{T} –matrix estimator is now checked by examining the effect of the correction on the RPSFs in a neighbouring region \mathcal{W}_2 that lies directly below the area \mathcal{W}_1 where we extract the aberration law (see new Fig.3c). Applying the aberration correction on a region different from where it was estimated allows us to avoid any overfitting bias. The following changes have been made in the manuscript:

- Page 12, Figure 3: New panels g and h have been included to show the evolution of the RPSF in \mathcal{W}_2 when phase laws displayed in panels e and f are used to compensate for aberrations.

The caption of Fig.3 has been modified accordingly.

- Page 10-11, lines 190-197: “The validity of the \mathbf{T} –matrix estimator in a region \mathcal{W}_1 (Fig. 3c) is investigated by examining the corrected RPSF in a neighbour region \mathcal{W}_2 (yellow box). \mathcal{W}_1 and \mathcal{W}_2 are sufficiently close to assume, in a first approximation, that they belong to the same isoplanatic patch. If the box is too small (Fig. 3d₁), our estimator has not converged yet and the correction is not valid, as shown by the degraded quality of the RPSF in \mathcal{W}_2 [Fig. 3h₁] compared with its initial value [Fig. 3g]. With sufficient spatial averaging [Fig. 3d₃], a valid aberration law can be extracted as shown by a corrected RPSF close to be only diffraction-limited [Fig. 3h₃].”

Following these changes, we have also modified the paragraph showing the scalar product $P_{\text{in/out}}$ (new notation) between $\hat{\mathbf{T}}_{\text{in}}$ and $\hat{\mathbf{T}}_{\text{out}}$ as a relevant observable for monitoring the convergence of the \mathbf{T} –matrix estimator:

- Page 11, lines 198-211: “The question that now arises is how we can, in practice, know if the convergence of $\hat{\mathbf{T}}$ is fulfilled without any *a priori* knowledge on \mathbf{T} . An answer can be found by comparing the estimated input and output aberration phase laws, $\hat{T}_{\text{in}}(\mathbf{u}, \mathbf{r}_p)$ and $\hat{T}_{\text{out}}(\mathbf{u}, \mathbf{r}_p)$, at a given point \mathbf{r}_p as shown in Figs. 3e and f. Spatial reciprocity implies that \hat{T}_{in} and \hat{T}_{out} shall be equal when the convergence of the estimator is reached [Figs. 3e₃,f₃]. Their normalized scalar product, $P_{\text{in/out}} = N_u^{-1} \hat{\mathbf{T}}_{\text{in}} \hat{\mathbf{T}}_{\text{out}}^\dagger$, can be used to probe the error made on the aberration phase law $|\delta T|^2$. Both quantities are actually related as follows (see Supplementary Section S7):

$$|\delta T|^2 \simeq 1 - P_{\text{in/out}}. \quad (3)$$

The normalized scalar product $P_{\text{in/out}}$ is displayed as a function of w and shows the convergence of the IPR process [Fig. 3a]. For a sufficiently large box [Fig. 3d₃], $\hat{\mathbf{T}}$ is supposed to have converged towards \mathbf{T} when $\hat{\mathbf{T}}_{\text{in}}$ and $\hat{\mathbf{T}}_{\text{out}}$ are almost equal [Fig. 3e₃,f₃], while, for a small box [Fig. 3d₁], a large discrepancy can be found between them. In the following, the parameter $P_{\text{in/out}}$ will thus be used as a guide star for monitoring the convergence of the UMI process.”

The impact of the imaging dimension on the convergence has also been developed to clarify our demonstration:

- Page 12, Figure 3a: The normalized scalar product $P_{\text{in/out}}$ (new notation) is now displayed as a function of the window size w , expressed in mm, instead of $N_{\mathcal{W}}$, expressed in resolution cells, to better illustrate the superiority of 3D over 2D imaging.
- Pages 11-12, lines 212-219: “The scaling law of Eq. 6 with respect to $N_{\mathcal{W}}$ is checked in Fig. 3b. The inverse scaling of the bias with $N_{\mathcal{W}}$ shows the advantage of 3D UMI over 2D UMI, since $N_{\mathcal{W}} \sim w^d$, with d the imaging dimension. This superiority is evident in Fig. 3a, which shows a faster convergence with 3D boxes (green curve) than with 2D patches (orange curve). For a given precision, 3D UMI thus provides a better spatial resolution for our \mathbf{T} –matrix estimator as shown by Figs. 3f₃ and f₄, where much better agreement between $\hat{\mathbf{T}}_{\text{in}}$ and $\hat{\mathbf{T}}_{\text{out}}$ is observed for a 3D box [Fig. 3d₃] than for a 2D patch [Fig. 3d₄] of same dimension w .”

6/ In Fig. 3 (and sometimes seen elsewhere, such as S2–S3), there is significant phase wrapping shown in the aberration phase laws. This wrapping is indicated by the transition from red-to-black-to-blue or blue-to-black-to-red. Phase wrapping normally creates issues with aberration correction.

Can the authors either explain why this method is not affected by this phase wrapping or how they unwrap the phase for correction?

We thank the referee for his relevant comment. Nevertheless, here, the focused reflection matrix is here a complex quantity and a simple phase conjugation of the aberration law is performed. Thus, an unwrapping of the phase would not change the result of the aberration correction process.

7/ Line 240-241: It is unclear what the authors were saying here (there could be a typo here, and I'm not sure what was trying to be said).

We just meant that our spatial reciprocity criterion $P_{\text{in/out}}$, the normalized scalar product between $\hat{\mathbf{T}}_{\text{in}}$ and $\hat{\mathbf{T}}_{\text{out}}$, is very low which is a manifestation of the imperfect convergence of our \mathbf{T} -matrix estimator. The corresponding sentence has been rephrased as follows:

- Page 14, lines 253-255: “The second difference is that our spatial reciprocity criterion $P_{\text{in/out}}$ is very low [see the blue boxplot in Fig. 4e]. This is the manifestation of a bad convergence of our \mathbf{T} -matrix estimator.”

8/ Line 312-313: What do you mean by more efficient? Supplementary section S3 did not describe efficiency.

Supplementary Figure S3 shows that the IPR algorithm allows us to evaluate the aberration law on a larger angular support than the first singular vector of the SVD. IPR is thus more efficient since it leads to a sharper compensation of aberrations. The problematic sentences have been rephrased as follows:

- Page 18, lines 325-328: “First, the proposed iterative phase reversal algorithm outperforms the SVD for local compensation of aberrations because it can evaluate the aberration law on a larger angular support (see Supplementary Figure S3), resulting in a sharper compensation of aberrations.”

9/ Line 315: "adaptive" seems out of place here. Did you mean to say local or individual aberration phase law?

Yes, that's right, the term “adaptive” was misleading and “local” would have been more appropriate. The corresponding paragraph has been fully rewritten following a previous remark of the referee (see our response to point 2).

10/ Line 327-328: I'm glad you included this and section S8 (and Fig. S7), because I think that there are consequences for "undersampled" transmissions, like that used in ultrafast imaging. Because the ultrafast transmissions are rather popular in research, it is important for readers to understand the implications here. I would probably also add that the "undersampled" plane wave transmissions will likely create a problem in having $\hat{\mathbf{T}}_{\text{in}}$ and $\hat{\mathbf{T}}_{\text{out}}$ converge since the underlying "PSF"s diverge significantly. In this paper, because the plane wave transmissions were highly sampled (1225 plane wave transmissions), there should be good agreement between the in and out "PSF"s.

We thank the reviewer for his comment, as we also consider this is an important aspect. The paragraph has been updated to discuss the problems encountered in UMI with ultrafast imaging:

(i) spatial aliasing that imposes a number of illuminations that scales with the aberration level and
(ii) symmetry breaking of reflection matrices, which can directly affect the accuracy of our spatial reciprocity criterion.

- Page 19, line 349-356: “However, due to spatial aliasing, the number of illuminations required for UMI scales with the number of resolution cells covered by the RPSF (see Supplementary Fig. S8). Because the aberration level through the skull is important, the illumination basis should thus be fully sampled. It limits 3D transcranial UMI to a compounded frame rate of only a few hertz, which is much too slow for ultrafast imaging [45]. Moreover, a reduced number of illuminations breaks the symmetry of the reflection matrix. It would therefore also affect the accuracy of our monitoring parameter based on spatial reciprocity.”

Line 433: "In the transducer basis ($o = k$),..." Should that say " $o = u$ " instead?

Yes, this is correct, the typo has been corrected (page 25, line 465).

Line 443: Is the assumption correct that D_{ri} means $D(\rho_{out}, o_{in}, z)$? If so, I find the notation here a bit hard to follow/read.

We apologize for the lack of clarity, there was actually a typo. The correction basis is now expressed with \mathbf{c} (new notation) instead of \mathbf{o} . According to this new notation, we meant the matrix $\mathbf{D}_{\mathbf{cr}} = [D(\mathbf{c}_{in}, \mathbf{r}_{out}) = D(\mathbf{c}_{in}, \{\boldsymbol{\rho}_{out}, z\})]$. This has been corrected in the new version of the manuscript:

- Page 26, lines 475-476: “ Note that the same operations can be performed by exchanging input and output to obtain the input distortion matrix $\mathbf{D}_{\mathbf{cr}} = [D(\mathbf{c}_{in}, \mathbf{r}_{out})] = [D(\mathbf{c}_{in}, \{\boldsymbol{\rho}_{out}, z\})]$.”
- To help the reader, matrix notations used in the paper are now summarized in Supplementary Table S2.

In the discussion of $N_{\mathcal{W}}$ as shown in Eq. 6, it was stated that the bias scales inversely proportional to $N_{\mathcal{W}}$. I was never able to fully follow the discussion in the paper and the supplementary material on that, but I did find (through the authors prior publications and not this one!) that reference [15] help greatly. In the discussions in the paper and supplementary material, this reference was not cited, but rather other citations that did not seem to address this topic.

Right, we now refer to this work (now Ref. [16]) when the scaling of the estimator bias with $N_{\mathcal{W}}$ is discussed:

- Page 10, line 183-185: “on the other hand, these areas should be large enough to encompass a sufficient number of independent realizations of disorder [16,19]”
- Supplementary Material, Page 4, lines 40-41 : “The intensity of the perturbation term scales as the inverse of the number $N_{\mathcal{W}} = (w_{\rho}^2 w_z) / (\delta \rho_0^2 \delta z_0)$ of resolution cells in each sub-region [1-3]”

C. Miscellaneous changes

For sake of clarity and rigor, some minor changes have been made to the notations:

- The coherence factor is now given as \mathcal{C} instead of C , since C also corresponds to the coefficients of the correlation matrix.
- The contrast is now given as \mathcal{F} instead of \mathcal{C} .
- The normalised scalar product used to evaluate the spatial reciprocity in aberration laws is now noted “ $P_{\text{in/out}}$ ”.
- $\Delta\tau$ denotes a time delay, used for example, to generate a plane wave, while τ denotes a time-of-flight used for beamforming, and t denotes absolute time.
- \mathbf{r} denotes the spatial coordinates of a point in the medium. \mathbf{r}_p denotes the center of a patch delimited by the window function \mathcal{W} . \mathbf{r}_m denotes the common midpoint between input and output focal spots.
- The propagation matrix is now referred to as \mathbf{G} instead of \mathbf{T}_0 to avoid any confusion with the transmission matrix \mathbf{T} that contains the aberration phase laws.
- \overline{RPSF} refers to the normalised RPSF to have simpler equations for the multiple scattering and the electronic noise rates $\alpha_{M/N}$ (Eqs. 31 & 32).
- $R_{\mathcal{M}}$ denotes the focused reflection matrix in a common midpoint representation.
- β is now an anti-symmetrical rate instead of a symmetrical rate.
- The bias intensity previously denoted by $\|\delta\hat{\mathbf{T}}\|^2$ has been replaced by $|\delta T|^2$.
- Notations have been updated in the flowchart of Fig. S1.
- Notations and symbols used in this manuscript are summarized in a set of tables at the end of the Supplementary Material.
- The correction basis is now expressed as $\mathbf{c}_{\text{in/out}}$ instead of $\mathbf{o}_{\text{in/out}}$, where the subscript “in” refers to the input and “out” to the output.

References

- [1] Mathieu Pernot, Mickaël Tanter, and Mathias Fink. 3-D real-time motion correction in high-intensity focused ultrasound therapy. Ultrasound Med. Biol., 30(9):1239–1249, September 2004.
- [2] Laura M. Hinkelman, Dong-Lai Liu, Leon A. Metlay, and Robert C. Waag. Measurements of ultrasonic pulse arrival time and energy level variations produced by propagation through abdominal wall. J. Acoust. Soc. Am., 95(1):530–541, January 1994.

REVIEWERS' COMMENTS

Reviewer #1 (Remarks to the Author):

All comments and concerns have been diligently addressed. The manuscript is, in my opinion, ready for publication.

Reviewer #2 (Remarks to the Author):

The authors did very well at responding to the reviewer's comment. I was puzzled, though, at the authors response regarding the phase wrapping comment. The author's remarked that "Thus, an unwrapping of the phase would not change the result of the aberration correction process." However, if the phase is not unwrapped, then the "extreme" delays could be shifted in the wrong direction. I believe this problem was common in the beamsum algorithm, although admittedly I could not recall or find any papers on the beamsum method that discuss this issue - this was mainly from my own personal experience. However, there are several papers that refer to this issue as wavelength or phase jumps. This was one of the better references that I could find that mention applying unwrapping of phase to counteract the issue:

Mozaffarzadeh, M.; Verweij, M.D.; de Jong, N.; Renaud, G. Comparison of Phase-Screen and Geometry-Based Phase Aberration Correction Techniques for Real-Time Transcranial Ultrasound Imaging. *Appl. Sci.* 2022, 12, 10183. <https://doi.org/10.3390/app121910183>

I'm not looking for the authors to change any of their methods, but I think it is worthy of looking into whether phase unwrapping is or is not relevant.

In the following we will provide a detailed response (in blue) to the reports of Referees (shown in italic black):

A. Introduction

We would like to thank the referees for their thoughtful feedback. Below we submit our answers to the referee comments and questions, and detail the changes made to the manuscript in response to the referee reports.

B. Response to Referee Reports

In the following, we answer the Referees' concerns and describe changes made to the manuscript in response to their comments. Referee comments are given in italics, followed by our responses in blue. Note that numbered references refer to the current (newly resubmitted) version of the manuscript.

Referee 1

All comments and concerns have been diligently addressed. The manuscript is, in my opinion, ready for publication.

We would like to express our gratitude for the thorough revision and constructive comments provided by the reviewer. Their feedback was extremely relevant and helped us to improve the quality and clarity of our research.

Referee 2

The authors did very well at responding to the reviewer's comment. I was puzzled, though, at the authors response regarding the phase wrapping comment. The author's remarked that "Thus, an unwrapping of the phase would not change the result of the aberration correction process." However, if the phase is not unwrapped, then the "extreme" delays could be shifted in the wrong direction. I believe this problem was common in the beamsum algorithm, although admittedly I could not recall or find any papers on the beamsum method that discuss this issue - this was mainly from my own personal experience. However, there are several papers that refer to this issue as wavelength or phase jumps. This was one of the better references that I could find that mention applying unwrapping of phase to counteract the issue:

Mozaffarzadeh, M.; Verweij, M.D.; de Jong, N.; Renaud, G. Comparison of Phase-Screen and Geometry-Based Phase Aberration Correction Techniques for Real-Time Transcranial Ultrasound Imaging. Appl. Sci. 2022, 12, 10183. <https://doi.org/10.3390/app121910183>

I'm not looking for the authors to change any of their methods, but I think it is worthy of looking into whether phase unwrapping is or is not relevant.

We thank the referee for his relevant comment on an issue that we probably underestimated in our previous response. In "conventional" phase aberration correction techniques, compensation of wave distortions is performed via application of time delays. So phase unwrapping is critical to properly

extract the time delay distribution (in particular for time delays larger than the mean period T of the wave-field). In our approach, compensation of wave distortions is applied to a complex focused reflection matrix by rephasing each angular component of the wave-field. Hence, as said in the previous round of review, a phase unwrapping would not change the result provided by our method.

However, we acknowledge that it does not mean that extreme time delays are not an issue. Indeed, the building of the broadband focused reflection matrix amounts to select singly-scattered echoes by means of coherent time gating. If fluctuations of time delays are much larger than the time resolution of our measurements ($\delta t \propto \Delta f^{-1}$), the different angular components of each echo will not emerge in the same time gate and the estimation of the focusing law will be imperfect. One manifestation of such extreme time delays would be a bad axial resolution for the ultrasound image. One way to circumvent this issue is to investigate the broadband focused reflection matrix over a set of reduced frequency bandwidths and tailor a more complex focusing law to address the problem of strong aberrations (inducing extreme time delay fluctuations), reverberations and multiple scattering. So yes, extreme time delays can be an issue but, in the present paper, we can judge *a posteriori* that we were not in this regime, given the satisfying axial resolution of the obtained ultrasound images.

Following the reviewer's suggestion, we now insist on this limit of UMI in the discussion part of the manuscript:

- Page 20, lines 378-383: “So far, one limit of UMI concerns the strong aberration regime in which extreme time delay fluctuations can occur. Indeed, our approach relies on a broadband focused reflection matrix that consists in a coherent time gating of singly-scattered echoes. If time delay fluctuations are larger than the time resolution δt of our measurement, the angular components of each echo will not necessarily emerge in the same time gate and aberration compensation will be imperfect.”

C. Miscellaneous changes

- Captions of Figs. 3, 4 and 5 have been detailed to describe more precisely each of their panels and sub-panels.